# Mace: A flexible framework for membership privacy estimation in generative models

**Yixi Xu**[*]                                                                          *yixx@microsoft.com*
*Microsoft*

**Sumit Mukherjee***                                                        *sumitmukherjee2@gmail.com*
*Insitro*

**Xiyang Liu**                                                                *xiyangl@cs.washington.edu*
*University of Washington*

**Shruti Tople**                                                               *shruti.tople@microsoft.com*
*Microsoft Research*

**Rahul Dodhia**                                                              *rahul.dodhia@microsoft.com*
*Microsoft*

**Juan Lavista Ferres**                                                        *jlavista@microsoft.com*
*Microsoft*

**Reviewed on OpenReview:** *https://openreview.net/forum?id=ZxmOkNe3u7*

## Abstract

Generative machine learning models are being increasingly viewed as a way to share sensitive data between institutions. While there has been work on developing differentially private generative modeling approaches, these approaches generally lead to sub-par sample quality, limiting their use in real world applications. Another line of work has focused on developing generative models which lead to higher quality samples but currently lack any formal privacy guarantees. In this work, we propose the first formal framework for membership privacy estimation in generative models. We formulate the membership privacy risk as a statistical divergence between training samples and hold-out samples, and propose sample-based methods to estimate this divergence. Compared to previous works, our framework makes more realistic and flexible assumptions. First, we offer a generalizable metric as an alternative to the accuracy metric (Yeom et al., 2018; Hayes et al., 2019) especially for imbalanced datasets. Second, we loosen the assumption of having full access to the underlying distribution from previous studies (Yeom et al., 2018; Jayaraman et al., 2020), and propose sample-based estimations with theoretical guarantees. Third, along with the population-level membership privacy risk estimation via the optimal membership advantage, we offer the individual-level estimation via the individual privacy risk. Fourth, our framework allows adversaries to access the trained model via a customized query, while prior works require specific attributes (Hayes et al., 2019; Chen et al., 2019; Hilprecht et al., 2019).

## 1 Introduction

The past decade has seen much progress in machine learning, largely due to the rapid growth in the number of large-scale datasets. However, concerns about the privacy of individuals being represented in datasets have led to a variety of regulations that made it increasingly difficult to share sensitive data across institutions

---

[*]equal contribution

especially in healthcare (Voigt & Von dem Bussche, 2017). Recent progress in the area of generative machine learning has made it possible to share synthetic data, which reflects the statistical properties of the original datasets (Georges-Filteau & Cirillo, 2020). Such synthetic data has been shown to allow for the development of downstream machine learning models with limited loss of performance compared to the original data (Rajotte et al., 2021). This has led to synthetic data sharing being increasingly viewed as a potentially privacy preserving alternative to sharing the raw data (Tom et al., 2020) between institutions.

Despite the appeal of using synthetic data generated by generative models as an alternative to traditional data sharing, recent work has shown common generative modeling approaches are often vulnerable to a variety of privacy attacks (Hilprecht et al., 2019; Hayes et al., 2019; Chen et al., 2019). This led to the development of differentially private generative modeling approaches which allow for the generation of synthetic data while providing strong formal privacy guarantees (Xie et al., 2018; Jordon et al., 2018). However, in the case of high dimensional datasets (such as images), such approaches have been shown to produce synthetic samples of very poor quality for any reasonable level of guaranteed privacy (Xie et al., 2018; Mukherjee et al., 2019). For example, Mukherjee et al. (2019) generated 50,000 synthetic CIFAR10 samples using the differentially private GAN (Xie et al., 2018) with a large privacy budget $\varepsilon = 100$. Then, a classifier was trained on the synthetic data, however the accuracy was below 20% when validated on the test set. It was initially assumed that the poor performance was the result of loose privacy accounting (leading to an overestimation of $\varepsilon$), but this assumption is recently challenged by a work indicating that moments accountant-based approaches lead to tight estimates of $\varepsilon$ (Nasr et al., 2021). Thus greatly reducing the possibility of developing a differentially private generative modeling approach that could generate samples good enough to train a strong ML model in practical high-dimensional data settings.

More recent work has focused on developing novel generative learning methods that have been empirically shown to be protected against certain types of privacy attacks, such as membership inference attacks (Mukherjee et al., 2019; Chen et al., 2021). Despite these advancements in empirically improving the privacy of generative models in a few settings, there is currently no approach to provide formal privacy certificates for such models. This in turn has limited the usability of such models in real-world applications where the complete lack of formal certificates would pose a problem with regulators. A promising line of related work has been in using membership inference attacks to audit the privacy of trained machine learning models (primarily discriminative models) with theoretical justifications (Yeom et al., 2018; Jayaraman et al., 2020). More specifically, these works estimate the membership privacy risk of a model against a specific adversary. As a result, it would be computationally expensive to estimate the maximum risk when there is a large group of adversaries, or even impossible given an infinite set of adversaries. As a comparison, MACE is able to estimate the maximum privacy risk via the Bayes optimal classifier. Another limitation is that these frameworks assume a full access to the underlying data distribution. For example, Yeom et al. (2018) used a dataset including 4819 patients who were prescribed warfarin, collected by the International Warfarin Pharmacogenetics Consortium to demonstrate their methodology. However, the method could hardly generalize if the population of interest includes all the patients who were prescribed warfarin instead of the specific 4819 patients. While the above case is more common in practice, it is usually not feasible to get full access to this kind of sensitive data. Thus, existing methods (Yeom et al., 2018; Jayaraman et al., 2020) are not applicable, and this leaves a gap between theory and practice. To overcome this restriction, MACE allows for not only a full access but also a limited access to the data via a simple random sample. Furthermore, MACE is able to provide consistent estimators of membership privacy risks at both individual and population level. This allows us to estimate the membership privacy risks of different subgroups, which is usually different as shown by Feldman (2020) on long-tailed distributions. Last but not least, much of this line of work has focused on auditing differentially private discriminative models. As a comparison, we focus on trained generative models, which may or may not be differentially private.

To motivate our paper, let us look at a real-world application scenario. *A clinical research institution wants to publicly release a medical imaging dataset to enable machine learning model development using the data. However, due to concerns about the personal health information (PHI) in the dataset, they look into synthetic data generation. Having identified a viable generative modeling approach, the institution is faced with a few questions prior to data/model release: i) should they release the synthetic dataset or the trained generative model?, ii) if they just release synthetic data, does it matter how many synthetic samples they release?, iii)*

*how vulnerable is the synthetic data or the trained model to membership inference attacks.* Currently, there is no answer to these questions, unless making a strong assumption of the adversaries i.e. only considering a few specific heuristic membership inference attacks (Hayes et al., 2019; Hilprecht et al., 2019), which is not realistic in practice.

In this paper, we begin to answer these questions through the development of a flexible statistical framework to measure the membership privacy risk in generative models (MACE: Membership privACy Estimation). Our framework is built on the formulation of the membership privacy risk (given a query access) as a statistical divergence between the distribution of training-set and non-training-set samples. We show the utility of our framework using many SOTA queries from the literature and some new ones against common computer vision as well as medical imaging datasets.

Our primary contributions are as follows:

- We develop a framework to estimate the maximum membership privacy risk against adversaries that have query access to the model. Our framework can not only estimate membership privacy risks that are defined as the accuracy of the membership inference attack as in (Yeom et al., 2018), but also those from a more general risk class (Koyejo et al., 2014). This gives the users flexibility to measure the ability of a membership inference attack to distinguish members from non-members from different angles, especially when the training set is a small part of the total available dataset. In addition, MACE is capable of estimating the membership privacy risk given any scalar or vector valued attributes from a learned model, while prior works (Hayes et al., 2019; Chen et al., 2019; Hilprecht et al., 2019) restrict to a set of specific attributes.

- Our framework is able to measure the individual-level membership privacy risk. This measures the risk of each individual sample against specific modes of membership inference attacks, allowing users to identify those high risk individuals and decide whether to exclude high risk samples and re-train their model.

- We loosen the assumption of having full access to the training and non-training set from previous studies (Yeom et al., 2018; Jayaraman et al., 2020), and extend to the case where only a simple random sample is feasible. Furthermore, we derive consistent estimators for both the maximum membership privacy risk and the individual-level membership privacy risk with theoretical justifications.

- We demonstrate the usability of MACE by experiments that analyze the membership privacy risks with regards to various query types and generative model architectures on three real-world datasets via the membership advantage and the individual privacy risk under both the accuracy-based and generalized metrics.

## 2 Background

In this section, we present a brief background on query functions, membership inference attacks, attack experiments, and the Bayes optimal classifier. Then we briefly discuss the limitations of current membership inference approaches. We assume readers already possess a general understanding of generative models and Differential Privacy, but provide a short background section in the Appendix for the sake of completeness.

### 2.1 Notation

We introduce notations that will be used in the rest of the paper.

- Let $z = (x, y) \in \mathcal{Z} = \mathcal{X} \times \mathcal{Y}$ be a data point from a data distribution $\mathcal{D}$. Note that $y$ is some extra information for generative models such as conditional GANs. In other words, $z = x \in \mathcal{X}$ for normal generative models.

- We assume an adversary $\mathcal{A}$ would have query access to a model via a query function $Q_S(\cdot) : \mathcal{X} \times \mathcal{Y} \to \mathbb{R}^q$, where $q$ is an integer. Examples of adversaries and query functions are given in Sections 2.3 and 2.2, respectively.

- Let $S \sim \mathcal{D}^n$ be an ordered list of $n$ points. It is referred to as *training set*, sampled from $\mathcal{D}$. We will assume the training set $S$ to be fixed in this paper.

- $z \sim S$ denotes uniformly sampling from a training set $S$. Also, $z \sim \mathcal{D} \backslash S$ denotes uniformly sampling from the data distribution not including the training set $S$, which is referred to as sampling from a *hold-out set*.

- For a set of samples $\{z_1, z_2, \cdots, z_N\}$, we define their associated membership labels as $\{m_1, m_2, \cdots, m_N\}$, where $m_i = 1$ if $z_i$ is in the training set and $m_i = -1$ otherwise for $i = 1, \cdots, N$.

- For a given condition $\mathcal{C}$, let $\mathbb{I}(\mathcal{C}) = 1$ if the condition $\mathcal{C}$ holds, otherwise 0.

## 2.2 Query functions

In this subsection, we introduce some representative query functions $Q_S(z)$ considered in or motivated by prior work. Following Chen et al. (2019); Hilprecht et al. (2019); Hayes et al. (2019), we divide our attack settings based on the accessibility of model components: (1) access only to generated synthetic data and (2) access to models.

### 2.2.1 Accessible synthetic datasets

In the common practice of synthetic data releasing, researchers or data providers may consider releasing only generated datasets or just the generator. However, prior works (Chen et al., 2019; Hilprecht et al., 2019) have shown releasing generator/synthetic datasets can cause privacy leakage. Specifically, for the case where a generative model is released, Chen et al. (2019) consider the following query function $Q_S(z) = \min_w L_2(z, G(w))$, where $G$ is the generator released to the public. Alternatively, Hilprecht et al. (2019) first generate a large synthetic dataset $g_1, g_2, \cdots, g_n$ using the generator and then use the following query function:

$$Q_S(z) = -\frac{1}{n} \sum_{i=1}^{n} \mathbb{I}(g_i \in U_\varepsilon(z)) \log d(g_i, z), \tag{1}$$

where $d$ is some distance metric and $U_\varepsilon$ is $\varepsilon$-ball defined on distance metric $d$.

Similar to these approaches, we assume that the generator memorizes the training data thus it generates synthetic dataset close to the training data. Under this assumption, if a sample x is closer to the synthetic dataset, it is more likely that x belongs to the training data. Hence for a sample z, we consider using the nearest neighbor distance to synthetic datasets as the query function:

$$Q_S(z) = \min_{j \in [n]} d(z, g_j), \tag{2}$$

where $d$ is a distance metric.

### 2.2.2 Accessible models

In this setting, we assume the adversary has query access to the model (the discriminator and the generator in the case of GANs). Such a situation commonly arises when researchers open source model parameters, or share model parameters insecurely.

For generative models, especially GANs, the most successful attack known (Hayes et al., 2019) assumes adversaries to access the model via the following query:

$$Q_S(z) = D(z), \tag{3}$$

where $D(z)$ is the output of the discriminator corresponding to input sample $z$. Intuitively, if a sample is in the training set, the discriminator would be more likely to output high values. While the adversary could

solely access the discriminator via the above query, we introduce a query below allowing accessing both the generator and the discriminator.

$$Q_S(z) = (D(z), \min_{j \in [n]} d(z, g_j)) \,. \tag{4}$$

This attack is a combination of attacks described in Equations 3 and 2.

While the discriminator score has been shown to be a very effective query for generative models with one discriminator, many recent privacy preserving generative modeling approaches often have multiple discriminators (Jordon et al., 2018; Mukherjee et al., 2019), with each discriminator being exposed to a part of the training dataset. Our previous query will not be useful in such situations. Here, we consider the recent work privGAN((Mukherjee et al., 2019) and present two queries (one single dimensional and one multi-dimensional). The single dimensional query used in (Mukherjee et al., 2019) is as follows:

$$Q_S(z) = \max_i D_i(z) \,, \tag{5}$$

where $D(z)$ is the output of the discriminator $i$ corresponding to input sample $z$. We propose a new multi-dimensional query which is stated as follows:

$$Q_S(z) = (D_1(z), \cdots, D_r(z)) \,, \tag{6}$$

For the purposes of demonstration, in this paper we use $r = 2$.

## 2.3 Membership inference attack adversaries

The goal of a membership inference attack (MIA) (Li et al., 2013; Shokri et al., 2017; Truex et al., 2018; Long et al., 2017), is to infer whether a sample $z$ is a part of the training set $S$. In our paper, we assume an MIA access the model via a query function $Q_S$. Thus, an MIA adversary is equivalent as a classifier given $Q_S(z)$ as the input to predict whether $z$ belongs to $S$.

In this paper, we focus on generative machine learning models (such as GANs). The study of MIAs against generative models is a relatively new research area. Hayes et al. (2019) first demonstrated MIAs against GANs. They propose: i) a black-box adversary that trains a shadow GAN model using the released synthetic data, ii) a white-box adversary that uses a threshold on the discriminator score of a released GAN model. Hilprecht et al. (2019) demonstrates a black-box MIA adversary that uses only the generator of the GANs (or synthetic samples) and operates by thresholding the L2 distance between the query sample and the closest synthetic sample. These existing works focus on the construction of a strong binary classifier as the MIA adversary, given different query functions. The details of these query functions have been discussed in Section 2.2.

## 2.4 The attack experiment

The membership privacy risk that arises from a query to the model is usually evaluated through a membership privacy experiment (Yeom et al., 2018; Jayaraman et al., 2020). The experiment assumes we have sampled a training set $S$ with size $|S| = n$ from the data distribution $\mathcal{D}$. Then a learning algorithm is trained on $S$, and an adversary would have access to the trained model through a query function $Q_S : \mathcal{Z} \to \mathcal{Q}$. To be specific, an adversary is provided with the query output of a randomly sampled point $z$ from either $S$ (with probability $p$) or $\mathcal{D}\backslash S$ (with probability $1 - p$). The adversary would then get a utility 1 if it guesses the membership correctly or an utility $-1$ otherwise.

## 2.5 The Bayes optimal classifier

Since membership identification is essentially a binary classification task, a membership adversary can then be seen simply as a binary classification model. Indeed, many existing papers on membership inference explicitly train binary classifiers for the purpose of membership inference (Shokri et al., 2017). The performance of such classifiers is often also used to empirically measure the membership privacy risk of different models

(Mukherjee et al., 2019). As the binary classifiers used in such papers are heuristically chosen, there is no guarantee that a better classification model does not exist for the task. The classifier that minimizes the expected error rate (maximizes accuracy) is called the Bayes optimal classifier (Devroye et al., 2013). Given, samples $(x, y) \in \mathcal{R}^d \times \{-1, 1\}$, the Bayes optimal classifier is:

$$C^{Bayes}(x) = \underset{r \in \{-1, 1\}}{\arg \max} \mathbb{P}(Y = r | X = x). \tag{7}$$

Note: the Bayes optimal classifier can only be approximated in practical scenarios, since we rely on estimates of $\mathbb{P}(Y = r | X = x)$.

While the Bayes optimal classifier in Equation 7 was originally designed to maximize the classification accuracy, Koyejo et al. (2014) has extended it to a family of the generalized metrics. We first define TP, FP, FN and TN as $\mathbb{P}(\mathcal{A}(X) = 1, Y = 1)$, $\mathbb{P}(\mathcal{A}(X) = 1, Y = -1)$, $\mathbb{P}(\mathcal{A}(X) = -1, Y = 1)$ and $\mathbb{P}(\mathcal{A}(X) = -1, Y = -1)$ respectively. Next, we show the definition of the generalized metrics as follows:

$$\ell(\mathcal{A}, \mathcal{P}) = \frac{a_0 + a_{11}\text{TP} + a_{10}\text{FP} + a_{01}\text{FN} + a_{00}\text{TN}}{b_0 + b_{11}\text{TP} + b_{10}\text{FP} + b_{01}\text{FN} + b_{00}\text{TN}}, \tag{8}$$

where $\mathcal{A}$ is the adversary, $\mathcal{P}$ is the distribution, $a_0$, $b_0$, $a_{ij}$ and $b_{ij}$ are pre-defined scalars for $i = 0, 1$ and $j = 0, 1$, $(X, Y) \sim \mathcal{P}$. The generalized metric can be used to represent several commonly used metrics such as accuracy, PPV, TPR, TNR, WA etc. (see Appendix). It is then demonstrated that the Bayes optimal classifier for this family of generalized metrics takes the forms:

$$\text{sgn}(\eta(x) - t_\ell) \text{ or } \text{sgn}(t_\ell - \eta(x)), \tag{9}$$

where $\eta(x) := \mathbb{P}(Y = 1 | \mathcal{A}(X) = \mathcal{A}(x))$ and $t_\ell \in (0, 1)$ is a constant depending on the metric $\ell$, when the marginal distribution of $X$ is absolutely continuous with respect to the dominating measure on $\mathcal{X}$ (Koyejo et al., 2014). It is worth noting that the Bayes optimal classifier for the generalized metric can only be approximated in practical scenarios, due to the lack of closed-form expressions of $\eta(x)$ and $t_\ell$.

### 2.6 Limitations of current membership inference approaches

There are several limitations in the existing literature on membership inference. First, most papers (Hayes et al., 2019; Hilprecht et al., 2019) focus on developing novel heuristic membership inference attacks, which are often limited in scope and can hardly be extended to another query. This is particularly problematic as much of these heuristic approaches cannot readily generalize to the generative modeling setting. Second, the current formal membership privacy estimation frameworks Yeom et al. (2018); Jayaraman et al. (2020) require a full access to the underlying data distribution, while this is not always possible in practice. Third, no paper has yet provided a rigorous approach to estimate the membership privacy risk at the individual level. Fourth, for most of the current membership inference methods (Hayes et al., 2019; Hilprecht et al., 2019) probability $p$ of Experiment 1 is usually set as 0.5 to form a balanced binary classification problem. However, in practice, $p$ is usually much smaller than 0.5, as pointed out in prior work (Jayaraman et al., 2020; Rezaei & Liu, 2020). In this work we seek to address all these issues.

## 3 The Membership Privacy Risk Quantification

In this section, we first introduce the membership advantage of a given adversary. Then, we define the optimal membership advantage as the maximum membership advantage. It is the maximum expected membership privacy risk of any adversary for the whole population. Furthermore, we present the optimal membership inference adversary. Finally, the individual privacy risk would be proposed to estimate the membership privacy risk at the individual level. The first subsection focuses on the accuracy-based metric, and the second subsection extends to the generalized metrics. We first propose an experiment formalizing membership inference attacks, adapted from Yeom et al. (2018).

**Experiment 1.** *Let $\mathcal{D}$ be the data distribution on $\mathcal{Z}$. We first have a fixed training set $S \sim \mathcal{D}^n$ with size $|S| = n$ and have a trained model. An adversary $\mathcal{A} : \mathcal{Q} \rightarrow \{-1, 1\}$ would access the trained model via the query function $Q_S : \mathcal{Z} \rightarrow \mathcal{Q}$. The membership experiment proceeds as follows:*

1. *Randomly sample $m \in \{-1, 1\}$ such that $m = 1$ with probability p.*

2. *If $m = 1$, then uniformly sample $z \sim S$; otherwise sample $z \sim \mathcal{D} \backslash S$ uniformly.*

Let $\mathcal{E}(\mathcal{D}, S, Q_S)$ denote the distribution of $(z, m)$ in Experiment 1.

### 3.1 For the accuracy-based metric

To define the optimal membership advantage, we first introduce the membership advantage of an adversary $\mathcal{A}$ given the query $Q_S$, as given in Definition 4 of Yeom et al. (2018). We define the membership advantage of the query $Q_S$ by an adversary $\mathcal{A}$ as the rescaled expected accuracy of the membership inference attack adversary $\mathcal{A}$. When $p = 0.5$, the membership advantage is equal to the difference between the adversary's true and false positive rates.

**Definition 1.** *The membership advantage of $Q_S$ by an adversary $\mathcal{A}$ is defined as*

$$\text{Adv}_p(Q_S, \mathcal{A}) = 2\mathbb{P}\left(\mathcal{A}(Q_S(z)) = m\right) - 1, \tag{10}$$

*where the probability is taken over the random sample $(z, m)$ drawn from $\mathcal{E}(\mathcal{D}, S, Q_S)$.*

If the adversary $\mathcal{A}$ is random guessing, $\text{Adv}_p(Q_S, \mathcal{A}) = 0$. If the adversary always gets the membership right, then $\text{Adv}_p(Q_S, \mathcal{A}) = 1$. Further, note that $\mathbb{P}(\mathcal{A}(Q_S(z)) = m)$ is the expected accuracy of the adversary's predictions. Hence this definition of membership advantage is directly related to accuracy. After introducing the membership advantage, we define the optimal membership advantage as the maximum membership advantage of all possible adversaries.

**Definition 2.** *The optimal membership advantage is defined as*

$$\text{Adv}_p(Q_S) = \max_{\mathcal{A}} \text{Adv}_p(Q_S, \mathcal{A}). \tag{11}$$

The following lemma obtains the optimal adversary for the accuracy-based metric.

**Lemma 1.** *Given the query $Q_S$, the data distribution $\mathcal{D}$, the training set $S$ and the prior probability p, the Bayes optimal classifier $\mathcal{A}^*$ maximizing membership advantage is given by*

$$\mathcal{A}^*(Q_S(z)) = \text{sgn}\left(\mathbb{P}\left(m = 1 | Q_S(z)\right) - \frac{1}{2}\right), \tag{12}$$

*where the probability is taken over the random sample $(z, m)$ drawn from $\mathcal{E}(\mathcal{D}, S, Q_S)$.*

*Furthermore, the optimal membership advantage can be re-written as*

$$\text{Adv}_p(Q_S) = \mathbb{E}_z\left[|f_p(z)|\right], \tag{13}$$

*where*

$$f_p(z) = \frac{\mathbb{P}(Q_S(z)|m = 1)p - \mathbb{P}(Q_S(z)|m = -1)(1 - p)}{\mathbb{P}(Q_S(z)|m = 1)p + \mathbb{P}(Q_S(z)|m = -1)(1 - p)}. \tag{14}$$

*Proof.* See the complete proof in Appendix D.1 □

Now we introduce the individual privacy risk at a sample $z_0$. It is proportional to the accuracy of the Bayesian optimal classifier $\mathcal{A}^*$ conditioning on $Q_S(z) = Q_S(z_0)$.

**Definition 3.** *The individual privacy risk of a sample $z_0$ for a query $Q_S$ under the accuracy-based metric is defined as:*

$$\text{AdvI}_p(Q_S, z_0) = 2\mathbb{P}[\mathcal{A}^*(Q_S(z)) = m | Q_S(z) = Q_S(z_0)] - 1,$$

*where the probabilities are taken over the random sample $(z, m)$ drawn from $\mathcal{E}(\mathcal{D}, S, Q_S)$.*

We then provide several convenient properties of the individual privacy risk $\text{AdvI}_p(Q_S, z_0)$ under the accuracy-based metric at a sample $z_0$. First, we show that it could be rewritten as $|f_p(z_0)|$, where $f_p$ is defined in Equation 14. This can be used in the following sections to derive a consistent estimator and confidence interval. Secondly, we connect the individual privacy risk to the optimal membership advantage. Thirdly, we show that $\text{AdvI}_p(Q_S, z_0)$ is proportional to the highest accuracy conditioning on $Q_S(z_0)$. Fourthly, we establish the connection between the optimal membership advantage, the individual privacy risk and Differential Privacy.

**Theorem 1.** *Let $Q_S$ be the query function and $z_0$ a fixed sample $\in \mathcal{Z}$.*

1. *The individual privacy risk at $z_0$ can be re-written as*

$$\text{AdvI}_p(Q_S, z_0) = |f_p(z_0)|$$

2. *The optimal membership advantage is the expectation of the individual privacy risk given the sample $z$ as a random variable.*

$$\text{Adv}_p(Q_S) = \mathbb{E}_z[\text{AdvI}_p(Q_S, z)] \tag{15}$$

3. *The Bayesian optimal classifier $\mathcal{A}^*$ is optimal at the individual-level, as*

$$\text{AdvI}_p(Q_S, z_0) = 2 \max_{\mathcal{A}} \mathbb{P}[\mathcal{A}(Q_S(z)) = m | Q_S(z) = Q_S(z_0)] - 1$$

4. *If a training algorithm is $\varepsilon$-differentially private, then for any choice of $z_0$, we have both the optimal membership advantage and the individual privacy risk bounded by a constant determined by $\varepsilon$:*

$$\text{Adv}_p(Q_S), \text{AdvI}_p(Q_S, z_0) \leq \max\left\{ \left| \tanh\left( \frac{\varepsilon + \lambda}{2} \right) \right|, \left| \tanh\left( \frac{-\varepsilon + \lambda}{2} \right) \right| \right\},$$

*where $\lambda := \log\left( \frac{p}{1-p} \right)$. When $p = 0.5$ and $\varepsilon = 1$, we have $\text{Adv}_p(Q_S), \text{AdvI}_p(Q_S, z_0) \leq 0.462$ for any $z_0 \in \mathcal{Z}$.*

*Proof.* See the complete proof in Appendix D.3 □

### 3.2 For the generalized metrics

As previously mentioned, membership privacy leakage is a highly imbalanced problem (Jayaraman et al., 2020; Rezaei & Liu, 2020). For example, the training set in a medical dataset may consist of data from the patients admitted to a clinical study with a particular health condition and the distribution $\mathcal{D}$ may represent data from all patients (in the world). Notably, previous works (Hayes et al., 2019; Hilprecht et al., 2019; Mukherjee et al., 2019) used metrics such as accuracy, precision and recall to measure the privacy risks even for the highly imbalanced setting where $p = 0.1$ (Hayes et al., 2019). Although these attacks result in high accuracy, precision or recall during the privacy evaluation stage, it has been shown to suffer from a high false positive rate(Rezaei & Liu, 2020), thus is less useful in practice.

To overcome these issues, prior work (Jayaraman & Evans, 2019; Jayaraman et al., 2020) proposes to use the positive predictive value (PPV), which is defined as the ratio of true members predicted among all the positive membership predictions made by an adversary $\mathcal{A}$. Here, we allow users even more flexibility by adopting the generalized metric defined in Equation 8. Through Experiment 1, we define the following metric to measure the membership privacy risk under the generalized metric.

**Definition 4.** *The membership advantage of an adversary $\mathcal{A}$ that has access to a trained model via a query $Q_S$ under the generalized metric $\ell$ is defined as*

$$\text{Adv}_{\ell,p}(Q_S, \mathcal{A}) = \ell(\mathcal{A}(Q_S(\cdot)), \mathcal{E}(\mathcal{D}, S, Q_S)),$$

*where $\mathcal{E}(\mathcal{D}, S, Q_S)$ is the distribution generated by Experiment 1.*

After introducing the membership advantage under the generalized metric $l$, we define the optimal membership advantage as the maximum membership advantage.

**Definition 5.** *The optimal membership advantage is defined as*

$$\mathrm{Adv}_{\ell,p}(Q_S) = \max_{\mathcal{A}} \mathrm{Adv}_{\ell,p}(Q_S, \mathcal{A}).$$

Similar to the accuracy-based metric, the optimal adversary for the generalized metric is the Bayes optimal classifier. While we mention that the exact function depends on the dataset and the metric, in some cases there exists a closed form solution for it.

**Lemma 2.** *Given the query $Q_S$, the data distribution $\mathcal{D}$, the training set $S$ and the prior probability $p$, if $b_{11} = b_{01}$ and $b_{10} = b_{00}$, then the Bayes optimal classifier $\mathcal{A}^*$ maximizing membership advantage under the generalized metric $l$ asymptotically is given by*

$$\mathcal{A}^*(Q_S(z)) = \mathrm{sgn}\left(\mathbb{P}\left(m = 1 | Q_S(z)\right) - t_\ell\right), \tag{16}$$

*where the probability is taken over the sample pair $(z, m)$ drawn from $\mathcal{E}(\mathcal{D}, S, Q_S)$ and $t_\ell = \frac{a_{00} - a_{10}}{a_{11} - a_{10} - a_{01} + a_{00}}$.*

*Proof.* See the complete proof in Appendix D.2 ◻

As a sanity check, for accuracy, we have $t_{\mathrm{ACC}} = \frac{1}{2}$. A wide range of performance measure under class imbalance settings can be seen as instances of the family of metrics under Lemma 2. For example, AM measure (Menon et al., 2013) defined as $\mathrm{AM} := 1/2 \, (\mathrm{TPR} + \mathrm{TNR})$ has optimal threshold $t_{\mathrm{AM}} = p$. As another example, TPR or recall has the optimal threshold $t_{\mathrm{TPR}} = 0$, which means in practice, we can always predict as positive to get the highest recall.

Similar to the case of the accuracy based metric, we define the individual privacy risk for the generalized metric as follows:

**Definition 6.** *The individual privacy risk of a sample $z_0$ for a query $Q_S$ under generalized metric $\ell$ is defined as*

$$\mathrm{AdvI}_{l,p}(Q_S, z_0) = \ell(\mathcal{A}^*(Q_S(\cdot)), \mathcal{E}(\mathcal{D}, S, Q_S | Q_S(z_0)),$$

*where $\mathcal{A}^*$ is the Bayes Optimal Adversary under the generalized metris and $\mathcal{E}(\mathcal{D}, S, Q_S | Q_S(z_0)$ is the distribution generated by Experiment 1 given $Q_S(z) = Q_S(z_0)$.*

**Remark 1.** *While we focus on a single query function in this section, the result can be easily extended to a set of queries $\{Q_S^1, Q_S^2, \cdot, Q_S^q\}$ by computing the maximum risk of all the queries $Q_S^i$, $i = 1, \cdots, q$.*

## 4 Estimation of the Individual Privacy Risk

In the ideal world, we have a full access to the data distribution $\mathcal{D}$. As a result, we could get the exact numbers to describe the individual privacy risk and the optimal membership advantage by Definition 2- 6. However, this could hardly be true in practice. For example, it is almost impossible to get access to the health records of all the patients around the world, while a simple random sample could be feasible. Previous studies (Yeom et al., 2018; Jayaraman et al., 2020) dealt with this by assuming the existence of a subset to fully represent a distribution. For instance, Jayaraman et al. (2020) used a Texas hospital data set consisting of 67,000 patient records as the data distribution to validate their framework. However, their framework could hardly be adapted to the case when the 67,000 patient records are only part of the population of interest.

To conquer this limitation, we extend to the situation when one can only access the training set $S$ and the data distribution $\mathcal{D}$ via a simple random sample. Moreover, we will provide a practical and principled approach to estimate the individual privacy risk for both the accuracy-based and the generalized metrics, In order to achieve this, we first propose an experiment. Note that this experiment would also be used to estimate the optimal membership advantage in the following section.

**Experiment 2.** *Let $\mathcal{D}$ be the data distribution on $\mathcal{Z}$. We first have a training set $S \sim \mathcal{D}^n$ with size $|S| = n$ and have a trained model. An adversary would access the trained model via the query function $Q_S : \mathcal{Z} \to \mathcal{Q}$. The ratio $p$ is the prior probability. The sampling process proceeds as follows:*

*1. Uniformly sample $N_1 = Np$ points $x_1, \cdots, x_{N_1}$ from $S$. Assign $z_i = x_i$ and $m_i = 1$, for $i = 1, \cdots, N_1$.*

*2. Uniformly sample $N_2 = (1-p)N$ points $y_1, \cdots, y_{N_2}$ from $\mathcal{D} \backslash S$. Assign $z_i = y_{i-N_1}$ and $m_i = -1$, for $i = N_1 + 1, \cdots, N_2$.*

*3. Create a consolidated set of samples $\{(z_i, m_i)\}_{i=1}^N$, where $m_i = -1$ if $i \le N_1$ and $m_i = 0$ if $i \ge N_1$.*

We assume that $Np$ and $N(1-p)$ are always integers for simplicity in this paper. But keep in mind that all the algorithms, lemmas and theorems could be easily extended to the case when this assumption does not hold.

### 4.1 For the accuracy-based metric

By Theorem 1, we have $\mathrm{AdvI}_p(Q_S, z) = |f_p(z)|$. Thus, it is sufficient to estimate $f_p(z)$ to estimate $\mathrm{AdvI}_p(Q_S, z)$. In the following sub-sections, we describe the construction of consistent estimators for $f_p(z)$.

#### 4.1.1 Discrete queries

When $Q_S(z) \in \mathcal{Q}$ is discrete, for a particular output $j \in \mathcal{Q}$, let $r_j = \mathbb{P}(Q_S(z) = j | m = 1)$ and $q_j = \mathbb{P}(Q_S(z) = j | m = -1)$. Frequency-based plug-in methods have been used empirically in similar settings(Mukherjee et al., 2019; Yaghini et al., 2019). We simply collect samples from Experiment 2 and plug-in the fraction to estimate $r_j$ and $q_j$. Then we account for the estimation error of this process by using a Clopper-Pearson confidence interval (Clopper & Pearson, 1934). We find the $\frac{\delta}{2}$-Clopper Pearson lower bound for $r_j$ denoted as $\hat{r}_{j,\text{lower}}$ and the $\frac{\delta}{2}$-Clopper Pearson upper bound for $r_j$ denoted as $\hat{r}_{j,\text{upper}}$. Finally we have the following theorem.

**Theorem 2.** *Let $\{X_1, \cdots, X_{N_1}, Y_1, \cdots, Y_{N_2}\}$ randomly sampled by Experiment 2. If $Q_S(z) = j$, then*

*1. A consistent estimator of $f_p(z)$ is given as follows:*

$$\frac{\hat{r}_j p - \hat{q}_j(1-p)}{\hat{r}_j p + \hat{q}_j(1-p)},$$

*where $\hat{r}_j = \frac{1}{N_1} \sum_{i=1}^{N_1} \mathbb{I}(Q_S(X_i) = j)$, and $\hat{q}_j = \frac{1}{N_2} \sum_{i=1}^{N_2} \mathbb{I}(Q_S(Y_i) = j)$.*

*2. The $(1-\delta)$-confidence interval $C_{1-\delta}(z)$ of $f_p(z)$ is*

$$\left[ \frac{\hat{r}_{j,\text{lower}} p - \hat{q}_{j,\text{upper}}(1-p)}{\hat{r}_{j,\text{lower}} p + \hat{q}_{j,\text{upper}}(1-p)}, \frac{\hat{r}_{j,\text{upper}} p - \hat{q}_{j,\text{lower}}(1-p)}{\hat{r}_{j,\text{upper}} p + \hat{q}_{j,\text{lower}}(1-p)} \right] .$$

*Proof.* See the complete proof in Appendix D.4 $\qquad\qquad\qquad\qquad\qquad\qquad\qquad\qquad\qquad\qquad\qquad \square$

#### 4.1.2 Continuous queries

Consider when $Q_S$ is a continuous query. Let $r(x) = \mathbb{P}(Q_S(z) = x | m = 1)$ and $q(x) = \mathbb{P}(Q_S(z) = x | m = -1)$. We first use Kernel Density Estimators (KDE) for both $r(x)$ and $q(x)$.

Recall that for samples $v_1, v_2, \cdots, v_N \in \mathbb{R}^d$ from an unknown distribution $\mathcal{R}$ defined by a probability density function $r$, a KDE with bandwidth $h$ and kernel function $K$ is given by

$$\frac{1}{Nh^d} \sum_{i=1}^N K\left(\frac{v - v_i}{h}\right) . \tag{17}$$

Additionally, we have a plug-in confidence interval for KDE (Chen, 2017) (see Lemma 4). Using this, we have a consistent estimator for $f_p(z)$ with a confidence interval to estimate the uncertainty.

**Theorem 3.** *Let $Q_S$ be a continuous query and $z \in \mathcal{Z}$. Let $\{X_1, \cdots, X_{N_1}, Y_1, \cdots, Y_{N_2}\}$ randomly sampled from Experiment 2.*

1. *A consistent estimator of $f_p(z)$ is given as follows:*

$$\frac{\widehat{r}_N(Q_S(z))p - \widehat{q}_N(Q_S(z))(1-p)}{\widehat{r}_N(Q_S(z))p + \widehat{q}_N(Q_S(z))(1-p)},$$

*where $\widehat{r}_N(Q_S(z)) = \frac{1}{N_1 h^d} \sum_{i=1}^{N_1} K\left(\frac{Q_S(z) - Q_S(X_i)}{h}\right)$, and $\widehat{q}_N(Q_S(z)) = \frac{1}{N_2 h^d} \sum_{i=1}^{N_2} K\left(\frac{Q_S(z) - Q_S(Y_i)}{h}\right)$.*

2. *The $(1-\delta)$-confidence interval $C_{1-\delta}(z)$ of $f_p(z)$ is*

$$\left[\frac{\widehat{r}_{\text{lower}}(Q_S(z))p - \widehat{q}_{\text{upper}}(Q_S(z))(1-p)}{\widehat{r}_{\text{lower}}(Q_S(z))p + \widehat{q}_{\text{upper}}(Q_S(z))(1-p)}, \frac{\widehat{r}_{\text{upper}}(Q_S(z))p - \widehat{q}_{\text{lower}}(Q_S(z))(1-p)}{\widehat{r}_{\text{upper}}(Q_S(z))p + \widehat{q}_{\text{lower}}(Q_S(z))(1-p)}\right],$$

*where $[\widehat{r}_{\text{lower}}, \widehat{r}_{\text{upper}}], [\widehat{q}_{\text{lower}}, \widehat{q}_{\text{upper}}]$ are $(1-\delta/2)$ confidence intervals of $r(Q_S(z))$ and $q(Q_S(z))$ respectively. Both confidence intervals are derived by Lemma 4.*

*Proof.* See the complete proof in Appendix D.5 □

### 4.2 For the generalized metrics

Similar to the case of the accuracy-based metric, we propose a consistent estimator to estimate the individual privacy risk for the generalized metric. Though there is not always a closed-form expression for the individual privacy risk for the generalized metric, we are able to provide an explicit formula given a few conditions. The following theorem provides a way to calculate the confidence interval for the generalized metric given a known $t_\ell$.

**Theorem 4.** *Let $c_1 = (a_0 + a_{01})$, $c_2 = (a_0 + a_{00})$, $c_3 = (a_{11} - a_{01})$, $c_4 = (a_{10} - a_{00})$, $d_1 = (b_0 + b_{01})$, $d_2 = (b_0 + b_{00})$, $d_3 = (b_{11} - b_{01})$, $d_4 = (b_{10} - b_{00})$, and $z_0$ be a sample $\in \mathcal{Z}$. Under the following conditions:*

1. *The Bayes optimal classifier $\mathcal{A}^*$ for the generalized metric $l$ can be written in the form of $\text{sgn}(\eta(x) - t_\ell)$, and $t_\ell$ is a known constant;*

2. *$\mathbb{P}(m = 1 | Q_S(z) = Q_S(z_0)) \neq t_\ell$;*

3. *$pd_1\mathbb{P}(Q_S(z_0)|m = 1) + d_2(1-p)\mathbb{P}(Q_S(z_0)|m = -1) + pd_3\mathbb{P}(Q_S(z_0)|m = 1)\mathbb{I}(\mathbb{P}(m = 1|Q_s(z_0)) > t_\ell) + d_4(1-p)\mathbb{P}(Q_S(z_0|m = -1)\mathbb{I}(\mathbb{P}(m = 1|Q_s(z_0)) > t_\ell) \neq 0$;*

*we have*

1. *A consistent estimator of $\text{AdvI}_{l,p}(Q_S, z_0)$ can be given as follows:*

$$\frac{c_1\hat{r}p + c_2\hat{q}(1-p) + c_3\hat{r}p\mathbb{I}((1-t_\ell)p\hat{r} > t_\ell(1-p)\hat{q}) + c_4\hat{q}(1-p)\mathbb{I}((1-t_\ell)p\hat{r} > t_\ell(1-p)\hat{q})}{d_1\hat{r}p + d_2\hat{q}(1-p) + d_3\hat{r}p\mathbb{I}((1-t_\ell)p\hat{r} > t_\ell(1-p)\hat{q}) + d_4\hat{q}(1-p)\mathbb{I}((1-t_\ell)p\hat{r} > t_\ell(1-p)\hat{q})}. \quad (18)$$

*If $\mathcal{Q}$ is discrete and $Q_S(z_0) = j$, then $\hat{r} = \hat{r}_j$ and $\hat{q} = \hat{q}_j$, as defined in Theorem 2. If $Q_S$ is a continuous query, then $\hat{r} = \hat{r}_N(Q_S(z_0))$ and $\hat{q} = \hat{q}_N(Q_S(z_0))$, as defined in Theorem 3*

2. *The $(1-\delta)$-confidence interval of $\text{AdvI}_{l,p}(Q_S, z_0)$ follows by plugging the $(1-\delta/2)$ confidence intervals of $p$ and $q$ – $[\hat{r}_{lower}, \hat{r}_{upper}]$ and $[\hat{q}_{lower}, [\hat{q}_{upper}]$ into Equation 18. In the case of discrete queries, $[\hat{r}_{\text{lower}}, \hat{r}_{\text{upper}}] = [\hat{r}_{\text{j,lower}}, \hat{r}_{\text{j,upper}}]$, where $Q_S(z_0) = j$, , as defined in Theorem 2. In the case of continuous queries, $[\hat{r}_{\text{lower}}, \hat{r}_{\text{upper}}] = [\widehat{r}_{\text{lower}}(Q_S(z_0)), \widehat{r}_{\text{upper}}(Q_S(z_0))]$, as defined in Theorem 3. $\hat{q}_{lower}$ and $\hat{q}_{upper}$ are defined in a similar manner.*

*Proof.* See the complete proof in Appendix D.6. □

A wide range of performance measures satisfy Condition 1. For example, AM measure (Menon et al., 2013) has the optimal threshold $t_{\text{AM}} = p$, and $TPR$ has the optimal threshold $t_{\text{TPR}} = 0$. Condition 2 can be waived if $c_3 = c_4 = d_3 = d_4 = 0$. Metrics such as accuracy, recall and specificity meet the above condition. Condition 3 assumes a nonzero expected value of the denominator in Equation 18. As for the confidence interval in 2, there could exist a closed-form expression if for example, $\text{AdvI}(Q_S, z_0)$ is a monotonic function in terms of $\mathbb{P}(Q_S(z_0))|m = 1)/\mathbb{P}(Q_S(z_0))|m = -1)$. Otherwise, an optimization problem would need to be solved, as outlined in 2.

## 5 Estimation of the Optimal Membership Advantage

In the above section, we proposed to estimate individual privacy risk for both the accuracy-based and generalized metrics with theoretical justifications. In this section, we further develop methods to estimate the optimal membership advantage $\text{Adv}_p(Q_S)$.

### 5.1 For the accuracy-based metric

We first propose a consistent estimator for the optimal membership advantage given a discrete query.

**Theorem 5.** *Let $X_1, \cdots, X_{N_1}$ be i.i.d. random variables drawn from $S$ and $Y_1, \cdots, Y_{N_2}$ be i.i.d. random variables drawn from $\mathcal{D} \backslash S$ in Experiment 2, where $N_1 = pN$ and $N_2 = (1-p)N$. Assume that $\mathcal{Q}$ is a finite set. Define*

$$W_N = W(X_1, \cdots, X_{N_1}, Y_1, \cdots, Y_{N_2})$$
$$= \sum_{j \in \mathcal{Q}} |\frac{p}{N_1} \sum_{i=1}^{N_1} \mathbb{I}(Q_S(X_i) = j) - \frac{1-p}{N_2} \sum_{i=1}^{N_2} \mathbb{I}(Q_S(Y_i) = j)| .$$

*Then (1) $W_N$ is a consistent estimator of the optimal membership advantage $\text{Adv}_p(Q_S)$; (2) $\mathbb{P}(|W_N - \mathbb{E}W_N| \geq \sqrt{\frac{2}{N} \log(\frac{2}{\delta})}) \leq \delta$.*

*Proof.* See the complete proof in Appendix D.7. $\square$

We then propose a consistent estimator for the optimal membership advantage given a continuous query.

**Theorem 6.** *Let $X_1, \cdots, X_{N_1}$ be i.i.d. random variables drawn from $S$, and $Y_1, \cdots, Y_{N_2}$ be i.i.d. random variables drawn from $\mathcal{D} \backslash S$ in Experiment 2, where $N_1 = Np$ and $N_2 = N(1-p)$. Assume that $\mathcal{Q} = \mathbb{R}^d$ and $h \to 0$, as $N \to \infty$. Define*

$$U_N = U(X_1, \cdots, X_{N_1}, Y_1, \cdots, Y_{N_2})$$
$$= \int |\frac{p}{N_1 h^d} \sum_{i=1}^{N_1} K(\frac{x - Q_S(X_i)}{h}) - \frac{1-p}{N_2 h^d} \sum_{i=1}^{N_2} K(\frac{x - Q_S(Y_i)}{h})| dx.$$

*Then (1) $U_N$ is a consistent estimator of the optimal membership advantage $\text{Adv}_p(Q_S)$; (2) $\mathbb{P}(|U_N - \mathbb{E}U_N| \geq \sqrt{\frac{2}{N} \log(\frac{2}{\delta})}) \leq \delta$.*

*Proof.* See the complete proof in Appendix D.8. $\square$

The practical estimation of the optimal membership advantage for the accuracy-based metric is described in Algorithm 1 using Monte Carlo integration.

---

**Algorithm 1** Practical estimation of the optimal membership advantage $\widehat{\mathrm{Adv}}_p(Q_S)$

---

**Input:** number of samples $N$, prior of membership $p$, training set $S$, query function $Q_S$, confidence level $\delta$
**Output:** the optimal membership advantage estimate $\widehat{\mathrm{Adv}}_p(Q_S)$
  1: Perform Experiment 2 and draw one set of samples $\{z_i\}_{i=1}^N$ with membership $\{m_i\}_{i=1}^N$ respectively.
  2: **if** $Q_S$ is discrete **then**
  3:     Use the tuples $\{(z_i, m_i)\}_{i=1}^N$ to calculate $W_N$
  4:     $\widehat{\mathrm{Adv}}_p(Q_S) \leftarrow W_N$
  5: **else**
  6:     Use the tuples $\{(z_i, m_i)\}_{i=1}^N$ to calculate $U_N$
  7:     $\widehat{\mathrm{Adv}}_p(Q_S) \leftarrow U_N$
  8: **end if**

---

### 5.2 For the generalized metrics

In this subsection, we propose consistent estimators for the optimal membership advantage when there exists a closed form solution for the optimal adversary.

Under some specific condition, Lemma 2 shows that the Bayes optimal classifier $\mathcal{A}^*$ maximizing membership advantage under the generalized metric $l$ is given by

$$\mathcal{A}^*(Q_S(z)) = \mathrm{sgn}\left(\eta(Q_S(z)) - t_\ell\right), \eta(Q_S(z)) = \mathbb{P}(m = 1 | Q_S(z)),\tag{19}$$

where $t_\ell = (a_{00} - a_{10})/(a_{11} - a_{10} - a_{01} + a_{00})$. Hence, we propose Algorithm 2 when the Bayes optimal classifier can be written as Equation 19 and $t_\ell$ is known. As shown in Algorithm 2, we first split the $N$ samples into two partitions, and use the first partition to obtain the estimator $\hat{\eta}(Q_S(z))$ for $\eta(Q_S(z))$. Following this, we estimate the Bayes optimal classifier $\mathcal{A}^*$ by

$$\hat{\mathcal{A}}(Q_S(z)) = \mathrm{sgn}\left(\hat{\eta}(Q_S(z)) - t_\ell\right).$$

Next, we use the second partition to obtain an empirical measure of $\mathrm{Adv}_{\ell,p}(Q_S, \hat{\mathcal{A}})$. More specifically, we

---

**Algorithm 2** Practical estimation of the optimal membership advantage $\mathrm{Adv}_{\ell,p}(Q_S)$ under a generalized metric $\ell$

---

**Input:** number of samples $N$, prior of membership $p$, training set $S$, query function $Q_S$, generalized metric $\ell$
**Output:** the empirical membership privacy $\mathrm{Adv}_{\ell,p}(Q_S)$ estimate
  1: Perform Experiment 2 and split the $N$ samples into two sets $S_1$ and $S_2$.
  2: Estimate $\eta(Q_S(z))$ by $\hat{\eta}(Q_S(z))$ using $S_1$.
  3: Let $\hat{\mathcal{A}}(Q_S(z)) = \mathrm{sgn}\left(\hat{\eta}(Q_S(z)) - t_\ell\right)$, where $t_\ell = (a_{00} - a_{10})/(a_{11} - a_{10} - a_{01} + a_{00})$.
  4: Calculate $\widehat{\mathrm{Adv}}_{\ell,p}(Q_S, \hat{\mathcal{A}})$ using $S_2$.

---

adopt the empirical measure defined by Koyejo et al. (2014). For any adversary $\mathcal{A}$, assuming that we sample $\{z_i, m_i\}_{i=1}^n$ by Experiment 2, we first define

$$\mathrm{TP}_n(\mathcal{A}) = \sum_{i=1}^n \frac{1}{n}\left(\frac{1}{2}\mathcal{A}(Q_S(z_i)) + \frac{1}{2}\right)\left(\frac{1}{2}m_i + \frac{1}{2}\right),\ \gamma_n(\mathcal{A}) = \frac{1}{n}\sum_{i=1}^n \left(\frac{1}{2}\mathcal{A}(Q_S(z_i)) + \frac{1}{2}\right)$$

as the empirical estimate of $\mathrm{TP} = \mathbb{P}(M = 1, \mathcal{A}(Q_S(Z)) = 1)$ and $\mathbb{P}(\mathcal{A}(Q_S(Z)) = 1)$. After this, we define the empirical measure of $\mathrm{Adv}_{\ell,p}(Q_S, \mathcal{A})$ as follows:

$$\widehat{\mathrm{Adv}}_{\ell,p}^n(Q_S, \mathcal{A}) = \frac{e_0 + e_1 \mathrm{TP}_n(\mathcal{A}) + e_2 \gamma_n(\mathcal{A})}{h_0 + h_1 \mathrm{TP}_n(\mathcal{A}) + h_2 \gamma_n(\mathcal{A})}\tag{20}$$

with the constants

$$e_0 = a_{01}p + a_{00} - a_{00}p + a_0, e_1 = a_{11} - a_{10} - a_{01} + a_{00}, e_2 = a_{10} - a_{00},$$

$$h_0 = b_{01}p + b_{00} - b_{00}p + b_0, h_1 = b_{11} - b_{10} - b_{01} + b_{00}, h_2 = b_{10} - b_{00}.$$

In Algorithm 2, the empirical measure $\widehat{\mathrm{Adv}}_{\ell,p}(Q_S, \hat{\mathcal{A}}) = \widehat{\mathrm{Adv}}_{\ell,p}^{|S_2|}(Q_S, \hat{\mathcal{A}})$.

Next, we introduce the following lemma giving the form of the Bayes optimal classifier when the query $Q_S$ is continuous.

**Lemma 3** (Koyejo et al. (2014)). *Assume that the marginal distribution of $Q_S(Z)$ in Experiment 1 is absolutely continuous with respect to the dominating measure on $Q_S(\mathcal{Z})$. Given the constants $e_0, e_1, e_2, h_0, h_1, h_2$ defined in Equation 20, define*

$$t_\ell^* = \frac{h_2 \mathrm{Adv}_{\ell,p}(Q_S) - e_2}{e_1 - h_1 \mathrm{Adv}_{\ell,p}(Q_S)}.$$

1. *When $e_1 > h_1 \mathrm{Adv}_{\ell,p}(Q_S)$, the Bayes optimal classifier $\mathcal{A}^*$ maximizing membership advantage under the generalized metric $\ell$ is given by $\mathrm{sgn}(\eta(Q_S(z)) - t_\ell^*)$;*

2. *When $e_1 < h_1 \mathrm{Adv}_{\ell,p}(Q_S)$, the Bayes optimal classifier $\mathcal{A}^*$ maximizing membership advantage under the generalized metric $\ell$ is given by $\mathrm{sgn}(t_\ell^* - \eta(Q_S(z)))$.*

By Lemma 3, the specific form of the Bayes optimal classifier relies on the unknown optimal membership advantage $\mathrm{Adv}_{\ell,p}(Q_S)$. Koyejo et al. (2014) suggested to estimate loose upper and lower bounds of $\mathrm{Adv}_{\ell,p}(Q_S)$ to determine the classifier. In the rest of this subsection, we assume $e_1 > h_1 \mathrm{Adv}_{\ell,p}(Q_S)$ so $\mathcal{A}^*(Q_S(z)) = \mathrm{sgn}(\eta(Q_S(z)) - t_\ell^*)$. The case where $e_1 < h_1 \mathrm{Adv}_{\ell,p}(Q_S)$ can be solved similarly.

Based on Lemma 3, we propose Algorithm 3 when the query $Q_S$ is continuous. In Algorithm 3, we first split the $N$ samples into three partitions, and use the first partition to obtain the estimator $\hat{\eta}(Q_S(z))$ for $\eta(Q_S(z))$. Next, we use the second partition to estimate $t_\ell$. Combing these two steps, we obtain the empirical Bayes optimal classifier $\hat{\mathcal{A}}(Q_S(z)) = \mathrm{sgn}(\hat{\eta}(Q_S(z)) - \hat{t}_\ell)$. We use the third partition to calculate the empirical measure of $\mathrm{Adv}_{\ell,p}(Q_S, \hat{\mathcal{A}})$ by Equation 20. Especially in Algorithm 3, $\widehat{\mathrm{Adv}}_{\ell,p}(Q_S, \hat{\mathcal{A}}) = \widehat{\mathrm{Adv}}_{\ell,p}^{|S_3|}(Q_S, \hat{\mathcal{A}})$.

---

**Algorithm 3** Practical estimation of the optimal membership advantage $\mathrm{Adv}_{\ell,p}(Q_S)$ under a generalized metric $\ell$ when $t_\ell$ is unknown

---

**Input:** number of samples $N$, prior of membership $p$, training set $S$, query function $Q_S$, generalized metric $\ell$
**Output:** the empirical membership privacy $\mathrm{Adv}_{\ell,p}(Q_S)$ estimate
 1: Perform Experiment 2 and draw three sets of samples $S_1$, $S_2$ and $S_3$.
 2: Estimate $\eta(Q_S(z))$ by $\hat{\eta}(Q_S(z))$ using $S_1$.
 3: Compute $\hat{t}_\ell = \arg\max_{x \in (0,1)} \widehat{\mathrm{Adv}}_{\ell,p}(Q_S, \mathrm{sgn}(\hat{\eta}(Q_S(\cdot)) - x))$ on $S_2$.
 4: Let $\hat{\mathcal{A}}(Q_S(z)) = \mathrm{sgn}(\hat{\eta}(Q_S(z)) - \hat{t}_\ell)$.
 5: Calculate $\widehat{\mathrm{Adv}}_{\ell,p}(Q_S, \hat{\mathcal{A}})$ using $S_3$.

---

We now introduce the following nice properties of the two algorithms. First, Theorem 7 shows that $\mathrm{Adv}_{\ell,p}(Q_S, \hat{\mathcal{A}})$ in Algorithm 2 is a consistent estimate of the optimal membership advantage $\mathrm{Adv}_{\ell,p}(Q_S)$ if $E|\hat{\eta} - \eta|^2 \to 0$. Using a suitable strongly proper loss function, we can obtain an estimator $\hat{\eta}$ satisfying $E|\hat{\eta} - \eta|^2 \to 0$ by the proof of Theorem 5 (Menon et al., 2013).

**Theorem 7.** *Assume that $b_{11} = b_{01}$, $b_{10} = b_{00}$, $a_{11} > a_{01}$ and $a_{00} > a_{10}$. Let $\hat{\mathcal{A}}$ outputted by Algorithm 2. If $\mathbb{E}_{Q_S(z)}(|\hat{\eta}(Q_S(z)) - \eta(Q_S(z))|^\sigma) \xrightarrow{p} 0$ for some $\sigma \geq 1$,*

$$\mathrm{Adv}_{\ell,p}(Q_S, \hat{\mathcal{A}}) \xrightarrow{p} \mathrm{Adv}_{\ell,p}(Q_S).$$

*Proof.* See the complete proof in Appendix D.9 □

Second, we show the consistency of the empirical measure of the membership advantage of any adversary by the proof of Lemma 8 (Koyejo et al., 2014).

**Theorem 8** (Koyejo et al. (2014)). *For each adversary $\mathcal{A}$, $\widehat{\mathrm{Adv}}_{\ell,p}^n(Q_S, \mathcal{A}) \xrightarrow{p} \mathrm{Adv}_{\ell,p}(Q_S, \mathcal{A})$.*

Finally, we show the following nice property of $\hat{t}_\ell$ - the estimate of $t_\ell$ in Algorithm 3.

**Theorem 9** (Koyejo et al. (2014)). *Assume that the marginal distribution of $Q_S(Z)$ in Experiment 1 is absolutely continuous with respect to the dominating measure on $Q_S(\mathcal{Z})$. Let $\hat{t}_\ell$ be outputted by Algorithm 3. If $\hat{\eta} \xrightarrow{p} \eta$,*

$$\text{Adv}_{\ell,p}(Q_S, \text{sgn}(\eta(Q_S(z)) - \hat{t}_\ell)) \xrightarrow{p} \text{Adv}_{\ell,p}(Q_S).$$

# 6 Experiments

In this section, we demonstrate how to use MACE by performing practical membership privacy estimation on trained generative models.

## 6.1 Setup

The different GAN architectures used were WGAN-GP (Gulrajani et al., 2017) (on CIFAR-10 & MNIST), JS-GAN (Goodfellow et al., 2014) (on skin-cancer MNIST) and privGAN (Mukherjee et al., 2019) (on MNIST). A JS-GAN is the original GAN formulation which uses a Jensen-Shannon divergence based loss. A WGAN-GP is an improved GAN formulation with a Wasserstein distance based loss with a gradient penalty. A privGAN is a GAN formulation that utilizes multiple generator-discriminator pairs and has been empirically shown to provide membership privacy.

Our experiments would be based on the following three real-world datasets: MNIST, CIFAR-10 and skin cancer MNIST. MNIST contains gray scale images of handwritten digits with 70000 digits from 0 to 9. CIFAR-10 comprises of 10 classes of 32 x32 RGB colored images with 60000 images in the whole dataset. Both of them are commonly used in the GAN literature. Additionally, to demonstrate a real world use case in healthcare, we use the skin cancer MNIST dataset (Tschandl et al., 2018) which comprises of 10,000 $64 \times 64$ RGB images of skin lesions (both cancerous and benign).

Following the common practice of membership inference attacks on generative models (Hayes et al., 2019; Mukherjee et al., 2019; Chen et al., 2019), we choose a random 10% subset of the entire dataset as a training set to show overfitting. In sub-section 6.3, 10% of the training images were corrupted by placing a white box at the center of images (with no changes to the non-training set images). For all the experiments, we set the confidence level $\delta = 0.05$. To create discrete queries, we bin the continuous interval into $100^d$ bins (where $d$ is the dimension of the query).

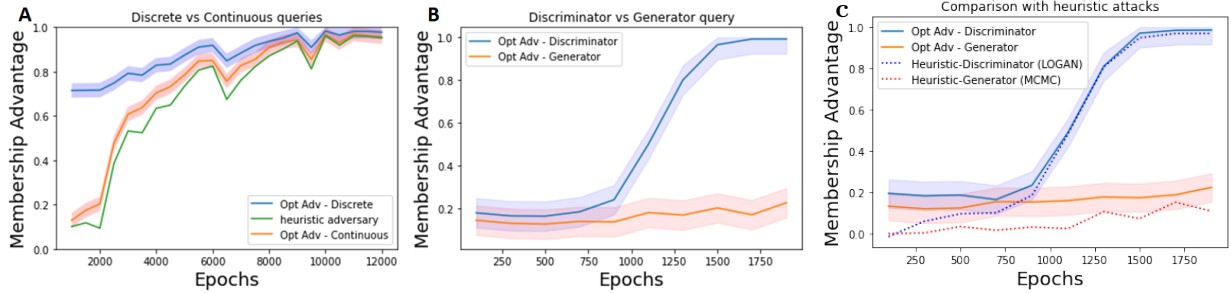

Figure 1: a) Comparison of discrete vs continuous queries against the WGAN-GP on the CIFAR-10 dataset. b) Comparison of queries against the generator and the discriminator on the skin-cancer MNIST dataset (for JS-GAN). In all cases $\delta$ is set to 0.05. c) Comparison of the optimal membership advantage and heuristic attacks' membership advantage for both queries against the generator and the discriminator on the skin-cancer MNIST dataset (for JS-GAN).

## 6.2 Estimation of the optimal membership advantage

### 6.2.1 For the accuracy-based metric

We first demonstrate the utility of our estimators of the optimal membership advantage under the accuracy-based metric with regard to different query types. In Figure 1a we show the applicability of our discrete and continuous estimators using the discriminator score as the query. Additionally, we compare the estimated performance of the optimal adversaries against a heuristic adversary that uses the same query (Hayes et al., 2019). We find that for both discrete and continuous queries, the estimated membership advantage is higher for the optimal adversary compared to the heuristic adversary. We note that the continuous query is seen to yield somewhat poorer performance than the discrete query, most likely due to sub-optimal selection of the KDE bandwidth. Optimal choice of the binning/KDE hyperparameters are beyond the scope of this paper but we direct readers to existing papers on this topic (Chen, 2015; Knuth, 2019). In Figure 1b we show the utility of our estimators on queries against accessible models and accessible datasets using a query of each type. We use the discriminator score as an example of queries on an accessible model and the query described in Equation 2 as an example of queries on an accessible synthetic dataset. Additionally, we compare our optimal membership advantage with the membership advantage of SOTA heuristic adversaries that use the similar queries as in (Hayes et al., 2019; Hilprecht et al., 2019). As widely reported (Hayes et al., 2019; Mukherjee et al., 2019), we find among our experiments that the optimal membership advantage is a lot smaller when adversaries gain access to the datasets compared to when they get direct access to the model. Furthermore, our optimal membership advantage estimates are higher than the SOTA heuristic adversaries in both settings. This validates Theorem 5 and 6, and demonstrates that our estimators are good estimators for the optimal membership advantage that would bound all the membership advantages including those due to the SOTA heuristic adversaries.

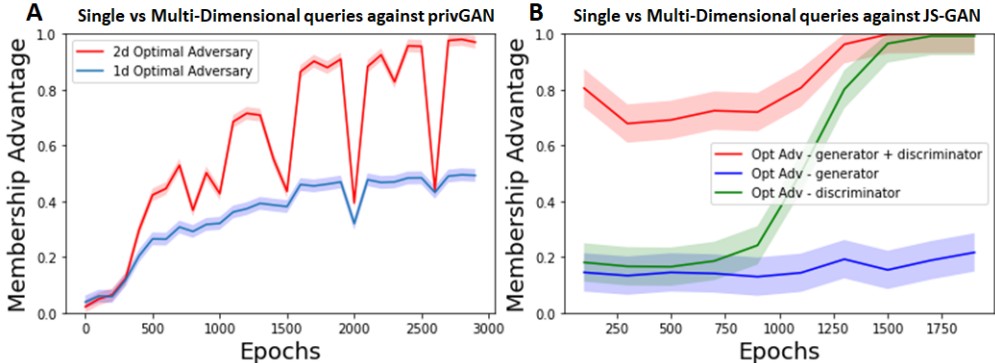

Figure 2: Comparison of single dimensional vs multi-dimensional queries against (a) privGAN on the MNIST dataset and (b) WGAN-GP on the skin-cancer MNIST dataset.

Next, we demonstrate the applicability of our estimators to multi-dimensional queries. In Figure 2a we compare the estimated membership advantage for a single and a multi-dimensional queries against privGAN (Mukherjee et al., 2019). We see that the estimated membership advantage with the multi-dimensional query is much higher than that of the 1-d query used in the privGAN paper. This indicates that while the privGAN is less likely to suffer from overfitting than the JS-GAN, releasing multiple discriminators could potentially increase it's privacy risk. In Figure 2b, we compare the estimated membership advantage for two single and one multi-dimensional queries against the JS-GAN. The multi-dimensional query is indeed a hybrid query that is the combination of the two 1-d queries (Equation 4). Intuitively, a hybrid query should impose a higher privacy risk than an individual query. Our experiment has validated this assumption and shown that the hybrid query has a higher estimated membership advantage than each individual 1-d query.

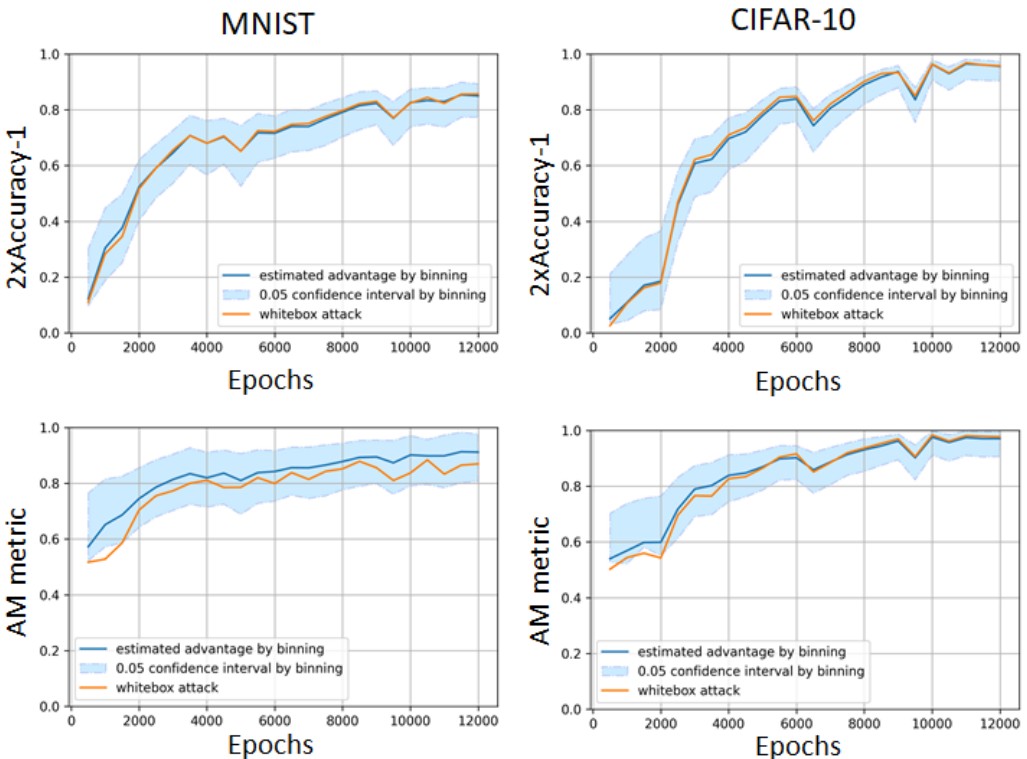

Figure 3: The membership advantages of the optimal adversary vs. the SOTA adversary against WGAN-GP under different metrics on the MNIST and CIFAR-10 datasets. For the sake of consistency, the membership advantage estimation for both metrics is done using the method used for the generalized metric.

### 6.2.2 Generalized metric

To demonstrate how to apply MACE under the generalized metrics, we qualitatively compare the membership advantages under AM and the accuracy-based metric. We set $p = 0.1$ here to construct an imbalanced dataset. In Figure 3, under both the accuracy-based metric and the AM metric, we compare the optimal membership advantage (the membership advantage of the optimal adversary) with the membership advantage of the heuristic adversary defined in (Hayes et al., 2019) using the discriminator score as the query. We find the heuristic adversary has comparable performance to the optimal adversary under both metrics. It is important to note here that while the optimal adversary is asymptotically optimal, on any set of samples there may be a stronger adversary possible.

### 6.3 Estimation of individual privacy risks

Having demonstrated the utility of our estimators for estimating the optimal membership advantage, here we demonstrate the utility of our estimators for the individual privacy risk. It is well known that samples from minority sub-groups can often be more vulnerable to membership inference attacks (Yaghini et al., 2019). Thus it is necessary to reveal the membership privacy at the individual level to reflect the true risks faced by the minority group. For the purposes of a simple demonstration, we constructed a toy dataset where 10% of the images were corrupted (described in section 6.1). After that, we trained a JS-GAN on this dataset and estimated the individual privacy risk of samples against the discriminator score query using the individual privacy risk estimator described in section 4.1.1. In Figure 4, we show (both qualitatively and quantitatively) that the corrupted images on average have a higher individual privacy risk than the uncorrupted images. This shows that outlier samples or certain sub-groups can be more vulnerable than the rest of the population, and the optimal membership advantage alone may not capture this information. It is worth noting here that there are particularly severe ramifications of higher privacy leakage risks of minorities in healthcare settings,

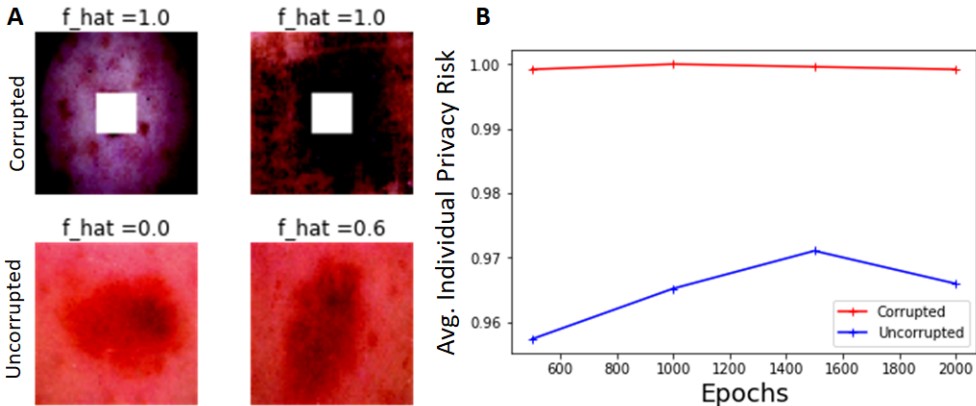

Figure 4: Application of individual privacy risk estimation. a) Example corrupted and uncorrupted images from skin-cancer MNIST, along with their estimated individual privacy risk. b) Comparison of average individual privacy risk between corrupted and uncorrupted images.

specially if such information points to the disease status. Data and model owners in such cases can consider retraining their generative model by omitting such sub-groups of samples.

## 7 Conclusion and Remarks

We developed the first formal framework that provides a certificate for the membership privacy risk of a trained generative model posed by adversaries having query access to the model at both the individual and population level. While our theory works regardless of the query dimension, we do not have a practical way to generate a meaningful certificate for the high-dimensional case. This would be a focus of our future work. Our framework works for a large family of metrics, allowing users flexible ways to measure the membership risk. Through experiments on multiple datasets, queries, model types and metrics, we show the practical applicability of our framework in measuring the optimal membership advantage as well as the individual privacy risk. Finally, to wrap up the paper, we re-visit our fictional example from the introduction to explain how MACE can help such data/model owners:

- By comparing the optimal membership advantage against a given trained model of queries that access the model via different practical queries, the model owners can determine the relative risk of releasing: i) the complete trained model, ii) parts of the trained model, iii) the synthetic dataset.

- For model owners who are only interested in releasing synthetic data, MACE can help identify how much data can be released for a desired level of membership advantage (see Appendix for example).

- For datasets containing sensitive groups of samples, MACE allows to estimate the group-level membership privacy risk through the estimation of the individual privacy risk and identify the most vulnerable minorities.

While we focus on generative models in this paper, our framework can be applied to discriminative models as well. An example application of our framework to a discrminative model is shown in the Appendix. While we focus on the theoretical aspects of membership privacy estimation in this paper, future work could look at using MACE to design new queries which can lead to stronger MIAs and a better understanding of membership privacy risks of different generative models. Another important direction is to derive a consistent estimator of the optimal membership advantage for the generalized metric by adapting Koyejo et al. (2014). It would also be interesting to extend our work to high-dimensional queries such as certain layers of the generator and discriminator.

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

## A  Background on Generative Adversarial Networks

Generative Adversarial Networks are the most common class of generative models. The original GAN algorithm (Goodfellow et al., 2014) learns a distribution of a dataset by adversarially training two modules, namely, a generator and a discriminator. The goal of the generator $G(w)$ is to learn a transformation that would convert a random vector $w$ to a realistic data sample. The goal of the discriminator module $D$ is to reliably distinguish synthetic samples (generated by the generator) from real samples. The mathematical formulation of this problem is as follows:

$$\min_G \max_D \mathbb{E}_{x \sim p_r(x)}[\log D(x)] + $$
$$\mathbb{E}_{x \sim p_G(x)}[\log(1 - D(x)] \ .$$

Here, $P_r$ is the real data distribution, and $P_G$ is the distribution of $G(w)$. There have been many GAN variants proposed since (Arjovsky et al., 2017; Mirza & Osindero, 2014; Berthelot et al., 2017). In this work, we examine our framework on the original GAN and some of its variations.

## B  Examples of common metrics that can be derived from the generalized metric

$$\text{ACC} = \frac{\text{TP} + \text{TN}}{\text{TP} + \text{FP} + \text{TN} + \text{FN}}$$
$$\text{PPV or Precision} = \frac{\text{TP}}{\text{TP} + \text{FP}}$$
$$\text{TPR or Recall} = \frac{\text{TP}}{\text{TP} + \text{FN}}$$
$$\text{TNR} = \frac{\text{TN}}{\text{FP} + \text{TN}}$$
$$\text{WA} = \frac{w_1\text{TP} + w_2\text{TN}}{w_1\text{TP} + w_2\text{TN} + w_3\text{FP} + w_4\text{FN}} \ .$$

## C  Background on differential privacy

The definition of $\varepsilon$-differential privacy is given as follows:

**Definition 7** ($\varepsilon$-Differential Privacy)**.** *We say a randomized algorithm $M$ is $\varepsilon$-differentially private if for any pair of neighbouring databases $S$ and $S'$ that differ by one record and any output event $E$, we have*

$$\mathbb{P}(M(S) \in E) \ \leq \ e^\varepsilon \mathbb{P}(M(S') \in E) \ . \tag{21}$$

# D    Detailed Proofs

## D.1    Proof for Lemma 1

*Proof.* As we can see from Definition 1, the membership advantage is defined as $2\text{Accuracy}(\mathcal{A}) - 1$. This means the Bayes classifier is given by $\text{sgn}\left(\mathbb{P}\left(m = 1|Q_S(z)\right) - \frac{1}{2}\right)$(Sablayrolles et al., 2019). We get the optimal membership advantage by plugging in this Bayes classifier. $\square$

## D.2    Proof for Lemma 2

*Proof.* When $b_{11} = b_{01}$ and $b_{10} = b_{00}$, our loss $\ell = \frac{a_0 + a_{11}\text{TP} + a_{10}\text{FP} + a_{01}\text{FN} + a_{00}\text{TN}}{b_0 + b_{11}(\text{TP}+\text{FN}) + b_{00}(\text{TN}+\text{FP})} = \frac{a_0 + a_{11}\text{TP} + a_{10}\text{FP} + a_{01}\text{FN} + a_{00}\text{TN}}{b_0 + b_{11}p + b_{00}(1-p)}$. It becomes a linear combination of TP, TN, FP and FN, which is also called cost sensitive classification as defined in (Elkan, 2001). As outlined in Elkan (2001), the optimal classifier would predict 1 when

$$\eta(Q_S(z))a_{01} + (1 - \eta(Q_S(z)))a_{00} \geq \eta(Q_S(z))a_{11} + (1 - \eta(Q_S(z)))a_{10} .$$

Rearranging the terms completes the proof. $\square$

## D.3    Proof for Theorem 1

*Proof.* By Equation 16 and Definition 3, we have $\text{AdvI}_p(Q_S, z_0) = |\mathbb{P}(m = 1|Q_S(z_0)) - \mathbb{P}(m = -1|Q_S(z_0))| = |f_p(z_0)|$, and 1 follows.

2 follows from Equation 15 and Equation 13.

Then, we prove 3. If $\mathcal{A}$ equals 1 at $Q_S(z_0)$, $2\mathbb{P}[\mathcal{A}(Q_S(z)) = m|Q_S(z) = Q_S(z_0)] - 1 = 2\mathbb{P}[m = 1|Q_S(z_0)] - 1 = \mathbb{P}[m = 1|Q_S(z_0)] - \mathbb{P}[m = -1|Q_S(z_0)]$. If $\mathcal{A}$ equals -1 at $Q_S(z_0)$, $2\mathbb{P}[\mathcal{A}(Q_S(z)) = m|Q_S(z) = Q_S(z_0)] - 1 = 2\mathbb{P}[m = -1|Q_S(z_0)] - 1 = \mathbb{P}[m = -1|Q_S(z_0)] - \mathbb{P}[m = 1|Q_S(z_0)]$. Thus, the maximum equals to $|\mathbb{P}(m = 1|Q_S(z_0)) - \mathbb{P}(m = -1|Q_S(z_0))| = \text{AdvI}_p(Q_S, z_0)$.

Finally we show 4. By the post-processing property, the $\varepsilon$-differential privacy indicates that the query output satisfies for any record z, we have

$$\left|\log \frac{\mathbb{P}(Q_S(z)|m = 1)}{\mathbb{P}(Q_S(z)|m = -1)}\right| \leq \varepsilon \tag{22}$$

Then 4 directly follows from 1, 2 and the fact that

$$\frac{x-1}{x+1} = \tanh(\frac{1}{2}\ln(x)). \tag{23}$$

$\square$

## D.4    Proof of Theorem 2

*Proof.* By the law of large numbers, we have $\hat{r}_j \xrightarrow{p} r_j$ and $\hat{q}_j \xrightarrow{p} q_j$. Then 1 follows from Slutsky's theorem (Corollary 1) and $pr_j + (1 - p)q_j > 0$. To prove 2, we first derive the $(1 - \delta/2)$ confidence intervals of $r_j$ and $q_j$ by Clopper & Pearson (1934). Then we could divide the nominator and the denominator by $q_j$, and 2 follows from the fact that $\frac{x-1}{x+1}$ is a monotonically increasing function for $x > 0$. $\square$

## D.5    Proof of Theorem 3

*Proof.* By Chen (2017), we have $\hat{r}_N(Q_S(z)) \xrightarrow{p} r(Q_S(z))$ and $\hat{q}_N(Q_S(z)) \xrightarrow{p} q(Q_S(z))$. Then 1 follows from Slutsky's theorem (Corollary 1) and $pr(Q_S(z)) + (1 - p)q(Q_S(z)) > 0$. To prove 2, we first derive the confidence intervals of $r(Q_S(z))$ and $q(Q_S(z))$ by Chen (2017). Then we could divide the nominator and the denominator by $q(Q_S(z))$, and 2 follows from the fact that $\frac{x-1}{x+1}$ is a monotonically increasing function for $x > 0$. $\square$

### D.6 Proof of Theorem 4

*Proof.* Define

$$r(Q_S(z_0)) = \mathbb{P}(Q_S(z_0)|m = 1), \ q(Q_S(z_0)) = \mathbb{P}(Q_S(z_0)|m = -1),$$

$$I(Q_S(z_0)) = \mathbb{I}\left(\mathbb{P}(m = 1|Q_S(z_0)) > t_\ell\right).$$

We will first show that

$$\mathrm{AdvI}_{l,p}(Q_S, z_0) = \frac{pc_1 r(Q_S(z_0)) + c_2(1-p)q(Q_S(z_0)) + pc_3 r(Q_S(z_0))I(Q_s(z_0)) + c_4(1-p)q(Q_S(z_0)I(Q_s(z_0))}{pd_1 r(Q_S(z_0)) + d_2(1-p)q(Q_S(z_0)) + pd_3 r(Q_S(z_0))I(Q_s(z_0)) + d_4(1-p)q(Q_S(z_0)I(Q_s(z_0))}. \tag{24}$$

Note that the individual privacy risk at a sample $z_0$ is written as

$$\mathrm{AdvI}_{l,p}(Q_S, z_0) = \frac{a_0 + a_{11}\mathrm{TP}(Q_S(z_0)) + a_{10}\mathrm{FP}(Q_S(z_0)) + a_{01}\mathrm{FN}(Q_S(z_0)) + a_{00}\mathrm{TN}(Q_S(z_0))}{b_0 + b_{11}\mathrm{TP} + b_{10}\mathrm{FP}(Q_S(z_0)) + b_{01}\mathrm{FN}(Q_S(z_0)) + b_{00}\mathrm{TN}(Q_S(z_0))}, \tag{25}$$

where $a_0$, $b_0$, $a_{ij}$ and $b_{ij}$ are pre-defined scalars for $i = 0, 1$ and $j = 0, 1$ as in Equation 8. $\mathrm{TP}(Q_S(z_0))$, $\mathrm{FP}(Q_S(z_0))$, $\mathrm{FN}(Q_S(z_0))$ and $\mathrm{TN}(Q_S(z_0))$ are the conditional versions of TP, FP, FN and TN on $Q_S(z) = Q_S(z_0)$. The four terms can be written as

$$\mathrm{TP}(Q_S(z_0)) = \mathbb{P}(\mathcal{A}^*(Q_S(z)) = 1, m = 1|Q_S(z_0)),$$

$$\mathrm{FP}(Q_S(z_0)) = \mathbb{P}(\mathcal{A}^*(Q_S(z)) = 1, m = -1|Q_S(z_0)),$$

$$\mathrm{FN}(Q_S(z_0)) = \mathbb{P}(\mathcal{A}^*(Q_S(z)) = -1, m = 1|Q_S(z_0)),$$

$$\mathrm{TN}(Q_S(z_0)) = \mathbb{P}(\mathcal{A}^*(Q_S(z)) = -1, m = -1|Q_S(z_0)).$$

By

$$\mathrm{FP}(Q_S(z_0)) = P(m = -1|Q_S(z_0)) - \mathrm{TN}(Q_S(z_0)) \ \& \ \mathrm{FN}(Q_S(z_0)) = P(m = 1|Q_S(z_0)) - \mathrm{TP}(Q_S(z_0)),$$

Equation 25 can be re-written as

$$\frac{a_0 + a_{10}\mathbb{P}(m = -1|Q_S(z_0)) + a_{01}\mathbb{P}(m = 1|Q_S(z_0)) + (a_{11} - a_{01})\mathrm{TP}(Q_S(z_0)) + (a_{00} - a_{10})\mathrm{TN}(Q_S(z_0))}{b_0 + b_{10}\mathbb{P}(m = -1|Q_S(z_0)) + b_{01}\mathbb{P}(m = 1|Q_S(z_0)) + (b_{11} - b_{01})\mathrm{TP}(Q_S(z_0)) + (b_{00} - b_{10})\mathrm{TN}(Q_S(z_0))}. \tag{26}$$

We first re-write

$$\mathrm{TP}(Q_S(z_0)) = \mathbb{P}(m = 1|Q_S(z_0))\mathbb{I}(\mathbb{P}(m = 1|Q_S(z_0)) > t_\ell)$$

and

$$\mathrm{TN}(Q_S(z_0)) = \mathbb{P}(m = -1|Q_S(z_0))\mathbb{I}(\mathbb{P}(m = 1|Q_S(z_0)) \le t_\ell)$$

by plug-in Equation 16. Then, by

$$\mathbb{P}(m = 1|Q_S(z_0)) = \frac{\mathbb{P}(Q_S(z) = Q_S(z_0)|m = 1)p}{\mathbb{P}(Q_S(z) = Q_S(z_0)|m = 1)p + \mathbb{P}(Q_S(z) = Q_S(z_0)|m = -1)(1-p)} \tag{27}$$

and

$$\mathbb{P}(m = -1|Q_S(z_0)) = \frac{\mathbb{P}(Q_S(z) = Q_S(z_0)|m = -1)(1-p)}{\mathbb{P}(Q_S(z) = Q_S(z_0)|m = 1)p + \mathbb{P}(Q_S(z) = Q_S(z_0)|m = -1)(1-p)},$$

we have Equation 26 equal to

$$\frac{pc_1 r(Q_S(z_0)) + c_2(1-p)q(Q_S(z_0)) + pc_3 r(Q_S(z_0))I(Q_s(z_0)) + c_4(1-p)q(Q_S(z_0)I(Q_s(z_0))}{pd_1 r(Q_S(z_0)) + d_2(1-p)q(Q_S(z_0)) + pd_3 r(Q_S(z_0))I(Q_s(z_0)) + d_4(1-p)q(Q_S(z_0)I(Q_s(z_0))}.$$

Hence, we have proved Equation 24.

In the next step, we will show the consistency of the proposed estimator. Note that, we have $\hat{r}(Q_S(z_0)) \xrightarrow{p} r(Q_S(z_0))$, $\hat{q}(Q_S(z_0)) \xrightarrow{p} q(Q_S(z_0))$ by the proof of Theorems 2 and 3. By Equation 27,

$$I(Q_S(z_0)) = \mathbb{I}\left(\frac{pr(Q_S(z_0))}{pr(Q_S(z_0)) + (1-p)q(Q_S(z_0))} > t_\ell\right) = \mathbb{I}((1-t_\ell)pr(Q_S(z_0)) > t_\ell(1-p)q(Q_S(z_0))).$$

It is a function of $r$ and $q$, and it is continuous except when $\mathbb{P}(m=1|Q_S(z_0)) = t_\ell$. By the second condition, we have $\mathbb{P}(m=1|Q_S(z_0)) \neq t_\ell$. Thus, $\mathbb{I}((1-t_\ell)p\hat{r} > t_l(1-p)\hat{q}) \xrightarrow{p} I(Q_S(z_0))$ follows from the continuous mapping theorem (Theorem 11).

By applying Slutsky's theorem (Corollary 1), we show that the nominator of Equation 18 converges in probability to the nominator in Equation 24. Similarly, we can prove that the denominator in Equation 18 converges in probability to the denominator in Equation 24. Because of the third condition, the denominator in Equation 24 is nonzero. Thus, 1 follows by Slutsky's theorem (Corollary 1).

Finally, we observe that given $(1-\delta/2)$-confidence intervals for $\mathbb{P}(Q_S(z)|m=1)$, $\mathbb{P}(Q_S(z)|m=-1)$, and the independence of $\mathbb{P}(Q_S(z)|m=1)$ and $\mathbb{P}(Q_S(z)|m=-1)$, the joint $(1-\delta)$-confidence interval of $\mathbb{P}(Q_S(z)|m=1)$ and $\mathbb{P}(Q_S(z)|m=-1)$ is simply the union of the two (using union bound), and 2 follows. $\qquad\square$

### D.7   Proof of Theorem 5

*Proof.* First note that

$$\mathcal{A}^*(Q_S) = \mathbb{E}_Z|f_p(Z)|$$
$$= \mathbb{E}_Z\left|\frac{\mathbb{P}(Q_S(Z)|M=1)p - \mathbb{P}(Q_S(Z)|M=-1)(1-p)}{\mathbb{P}(Q_S(Z))}\right|$$
$$= \sum_{j\in\mathcal{Q}}|\mathbb{P}(Q_S(Z)=j|M=1)p - \mathbb{P}(Q_S(Z)=j|M=-1)(1-p)|$$

It is sufficient to show that $\forall j \in \mathcal{Q}$,

$$\left|\frac{p}{N_1}\sum_{i=1}^{N_1}\mathbb{I}(Q_S(X_i)=j) - \frac{1-p}{N_2}\sum_{i=1}^{N_2}\mathbb{I}(Q_S(Y_i)=j)\right| \xrightarrow{p} |\mathbb{P}(Q_S(Z)=j|M=1)p - \mathbb{P}(Q_S(Z)=j|M=-1)(1-p)|,$$

$$(28)$$

and then (1) follows by Slutsky's Theorem (Corollary 1).

To prove Equation (28), we first show that

$$\frac{1}{N_1}\sum_{i=1}^{N_1}\mathbb{I}(Q_S(X_i)=j) \xrightarrow{p} \mathbb{P}(Q_S(Z)=j|M=1),$$

$$\frac{1}{N_2}\sum_{i=1}^{N_2}\mathbb{I}(Q_S(Y_i)=j) \xrightarrow{p} \mathbb{P}(Q_S(Z)=j|M=-1)$$

by the law of large numbers. Then Equation (28) follows by applying the continuous mapping theorem (Theorem 11).

Next, we prove (2). It is easy to verify that $\forall i \in \{1,\cdots,N_1\}$, $\forall x_1,\cdots,x_i,\cdots,x_{N_1},y_1,\cdots,y_{N_2},x_i' \in \mathcal{Z}$, we always have

$$|W(x_1,\cdots,x_i,\cdots,x_{N_1},y_1,\cdots,y_{N_2}) - W(x_1,\cdots,x_i',\cdots,x_{N_1},y_1,\cdots,y_{N_2})| \leq \frac{2p}{N_1} = \frac{2}{N}.$$

Similarly, for $\forall i \in \{1,\cdots,N_2\}$ and

$\forall x_1,\cdots,x_{N_1},y_1,\cdots,y_i,\cdots,y_{N_2},y_i' \in \mathcal{Z}$, we always have

$$|W(x_1, \cdots, x_{N_1}, y_1, \cdots, y_i, \cdots, y_{N_2}) - W(x_1, \cdots, x_{N_1}, y_1, \cdots, y_i', \cdots, y_{N_2})| \leq \frac{2(1-p)}{N_2} = \frac{2}{N}.$$

Then by McDiarmid's inequality (Theorem 13), $\mathbb{P}(|W_N - \mathbb{E}W_N| \geq t) \leq 2\exp\left(-\frac{N}{2}t^2\right)$. Let $\delta = 2\exp\left(-\frac{N}{2}t^2\right)$, and we have (2). $\qquad\square$

### D.8 Proof of Theorem 6

*Proof.* (1) We first prove the consistency. Note that

$$
\begin{aligned}
\mathcal{A}^*(Q_S) \quad &= \mathbb{E}_Z|f_p(Z)| \\
&= \mathbb{E}_Z \left| \frac{\mathbb{P}(Q_S(Z)|M=1)p - \mathbb{P}(Q_S(Z)|M=-1)(1-p)}{\mathbb{P}(Q_S(Z))} \right| \\
&= \int |r(x)p - q(x)(1-p)| \, dx,
\end{aligned}
$$

where $r(x) = \mathbb{P}(Q_S(z) = x|m = 1)$ and $q(x) = \mathbb{P}(Q_S(z) = x|m = -1)$. Rewrite $U_N$ as

$$
\begin{aligned}
U_N &= \int |\frac{p}{N_1 h^d} \sum_{i=1}^{N_1} K(\frac{x - Q_S(X_i)}{h}) - \frac{1-p}{N_2 h^d} \sum_{i=1}^{N_2} K(\frac{x - Q_S(Y_i)}{h})| dx \\
&= \int \left| p\left( \frac{1}{N_1 h^d} \sum_{i=1}^{N_1} K(\frac{x - Q_S(X_i)}{h}) - r(x) \right) + (pr(x) - (1-p)q(x)) + (1-p)\left( q(x) - \frac{1}{N_2 h^d} \sum_{i=1}^{N_2} K(\frac{x - Q_S(Y_i)}{h}) \right) \right| dx.
\end{aligned}
$$

Then

$$
\begin{aligned}
|U_N - \mathcal{A}^*(Q_S)| &\leq \int \left| p\left( \frac{1}{N_1 h^d} \sum_{i=1}^{N_1} K(\frac{x - Q_S(X_i)}{h}) - r(x) \right) + (1-p)\left( q(x) - \frac{1}{N_2 h^d} \sum_{i=1}^{N_2} K(\frac{x - Q_S(Y_i)}{h}) \right) \right| dx \\
&\leq p \int \left| \left( \frac{1}{N_1 h^d} \sum_{i=1}^{N_1} K(\frac{x - Q_S(X_i)}{h}) - r(x) \right) \right| dx + (1-p) \int \left| \left( q(x) - \frac{1}{N_2 h^d} \sum_{i=1}^{N_2} K(\frac{x - Q_S(Y_i)}{h}) \right) \right| dx \\
&\xrightarrow{p} 0,
\end{aligned}
$$

where the last step follows from Theorem 12 and Slutsky's theorem (Corollary 1).

(2) It is easy to verify that $\forall i \in \{1, \cdots, N_1\}$,
$\forall x_1, \cdots, x_i, \cdots, x_{N_1}, y_1, \cdots, y_{N_2}, x_i' \in \mathcal{Z}$, we always have

$$|U(x_1, \cdots, x_i, \cdots, x_{N_1}, y_1, \cdots, y_{N_2}) - U(x_1, \cdots, x_i', \cdots, x_{N_1}, y_1, \cdots, y_{N_2})| \leq \frac{2p}{N_1} = \frac{2}{N}.$$

Similarly, for $\forall i \in \{1, \cdots, N_2\}$ and
$\forall x_1, \cdots, x_{N_1}, y_1, \cdots, y_i, \cdots, y_{N_2}, y_i' \in \mathcal{Z}$, we always have

$$|U(x_1, \cdots, x_{N_1}, y_1, \cdots, y_i, \cdots, y_{N_2}) - U(x_1, \cdots, x_{N_1}, y_1, \cdots, y_i', \cdots, y_{N_2})| \leq \frac{2(1-p)}{N_2} = \frac{2}{N}.$$

Then by McDiarmid's inequality (Theorem 13), $\mathbb{P}(|U_N - \mathbb{E}U_N| \geq t) \leq 2\exp\left(-\frac{N}{2}t^2\right)$. Let $\delta = 2\exp\left(-\frac{N}{2}t^2\right)$, and we have (2). $\qquad\square$

### D.9   Proof of Theorem 7

*Proof.* First, we would bound $\mathrm{Adv}_{\ell,p}(Q_S) - \mathrm{Adv}_{\ell,p}(Q_S, \hat{\mathcal{A}})$. By the definition of $\mathcal{E}(\mathcal{D}, S, Q_S)$, we have

$$\mathbb{P}(m = 1, \mathcal{A}(Q_S(z)) = 1) + \mathbb{P}(m = 1, \mathcal{A}(Q_S(z)) = -1) = p$$

and

$$\mathbb{P}(m = -1, \mathcal{A}(Q_S(z)) = 1) + \mathbb{P}(m = -1, \mathcal{A}(Q_S(z)) = -1) = 1 - p.$$

Thus, we could rewrite the membership advantage under the generalized metric $\mathrm{Adv}_{\ell,p}(Q_S, \mathcal{A})$ as follows:

$$\frac{a_0 + a_{11}p + a_{00}(1-p) - (a_{11} - a_{01})\mathbb{P}(m = 1, \mathcal{A}(Q_S(z)) = -1) - (a_{00} - a_{10})\mathbb{P}(m = -1, \mathcal{A}(Q_S(z)) = 1)}{b_0 + b_{11}p + b_{00}(1-p)}.$$

Define $c = (a_{00} - a_{10})/(a_{00} - a_{10} + a_{11} - a_{01})$, $\beta_0 = (a_0 + a_{11}p + a_{00}(1-p))/(b_0 + b_{11}p + b_{00}(1-p))$, and $\beta_1 = (a_{00} - a_{10} + a_{11} - a_{01})/(b_0 + b_{11}p + b_{00}(1-p))$. We have

$$\mathrm{Adv}_{\ell,p}(Q_S, \mathcal{A}) = \beta_0 - \beta_1 \left[c\mathbb{P}(m = -1, \mathcal{A}(Q_S(z)) = 1) + (1-c)\mathbb{P}(m = 1, \mathcal{A}(Q_S(z)) = -1)\right]. \tag{29}$$

Because $a_{00} > a_{10}$ and $a_{11} > a_{01}$, $c \in (0, 1)$ and $\beta_1 > 0$. Define $\Theta = \{\mathcal{A} : \mathcal{A}(Q_S(z)) = \mathrm{sgn}\left(\phi(Q_S(z)) - t_\ell\right), \phi : \mathbb{R} \to [0, 1]\}$. Next, we would bound $\mathrm{Adv}_{\ell,p}(Q_S) - \mathrm{Adv}_{\ell,p}(Q_S, \hat{\mathcal{A}})$.

$$\mathrm{Adv}_{\ell,p}(Q_S) - \mathrm{Adv}_{\ell,p}(Q_S, \hat{\mathcal{A}}) = \max_{\mathcal{A}} \mathrm{Adv}_{\ell,p}(Q_S, \mathcal{A}) - \mathrm{Adv}_{\ell,p}(Q_S, \hat{\mathcal{A}})$$

$$= \max_{\mathcal{A} \in \Theta} \mathrm{Adv}_{\ell,p}(Q_S, \mathcal{A}) - \mathrm{Adv}_{\ell,p}(Q_S, \hat{\mathcal{A}}) \tag{30a}$$

$$= \beta_1 \left[c\mathbb{P}(m = -1, \hat{\mathcal{A}}(Q_S(z)) = 1) + (1-c)\mathbb{P}(m = 1, \hat{\mathcal{A}}(Q_S(z)) = -1)\right] -$$

$$\beta_1 \inf_{\mathcal{A}} \left[c\mathbb{P}(m = -1, \mathcal{A}(Q_S(z)) = 1) + (1-c)\mathbb{P}(m = 1, \mathcal{A}(Q_S(z)) = -1)\right] \tag{30b}$$

$$\leq \beta_1 \mathbb{E}_{Q_S(z)}(|\hat{\eta}(Q_S(z)) - \eta(Q_S(z))|^\sigma) \xrightarrow{p} 0. \tag{30c}$$

The step in Equation 30a follows from Lemma 2. The step in Equation 30b follows from Equation 29 and $\beta_1 > 0$. The step in Equation 30c follows from $\beta_1 > 0$, $c \in (0, 1)$ and Lemma 4 (Menon et al., 2013). Note that $\mathrm{Adv}_{\ell,p}(Q_S, \hat{\mathcal{A}}) \leq \mathrm{Adv}_{\ell,p}(Q_S)$. Hence we have $\mathrm{Adv}_{\ell,p}(Q_S, \hat{\mathcal{A}}) \xrightarrow{p} \mathrm{Adv}_{\ell,p}(Q_S)$.

$\square$

## E   Model architectures and hyper–parameters

Here we outline the different layers used in the model architectures for different datasets. The last layers of discriminators for WGAN experiments do not have sigmoid activation functions. The hyper-parameters are chosen the same same as (Goodfellow et al., 2014; Gulrajani et al., 2017).

### E.1   MNIST

#### E.1.1   Generator layers

- Dense(units= 512, input size= 100)
- LeakyReLU($\alpha = 0.2$)
- Dense(units= 512)
- LeakyReLU($\alpha = 0.2$)
- Dense(units= 1024)
- LeakyReLU($\alpha = 0.2$)
- Dense(units= 784, activation = 'tanh')

### E.1.2 Discriminator layers

- Dense(units= 2048)
- LeakyReLU($\alpha = 0.2$)
- Dense(units= 512)
- LeakyReLU($\alpha = 0.2$)
- Dense(units= 256)
- LeakyReLU($\alpha = 0.2$)
- Dense(units= 1, activation = 'sigmoid')

## E.2 CIFAR–10

### E.2.1 Generator layers

- Dense(units=$2 \times 2 \times 512$)
- Reshape(target shape= $(2, 2, 512)$)
- Conv2DTranspose(filters= 128, kernel size= 4, strides= 1)
- ReLU()
- Conv2DTranspose(filters= 64, kernel size= 4, strides= 2, padding= 1)
- ReLU()
- Conv2DTranspose(filters= 32, kernel size= 4, strides= 2, padding= 1)
- ReLU()
- Conv2DTranspose(filters= 3, kernel size= 4, strides= 2,padding= 1, activation = 'tanh')

### E.2.2 Discriminator layers

- Conv2D(filters= 64, kernel size= 5, strides= 2)
- Conv2D(filters= 128, kernel size= 5, strides= 2)
- LeakyReLU($\alpha = 0.2$)
- Conv2D(filters= 128, kernel size= 5, strides= 2)
- LeakyReLU($\alpha = 0.2$)
- Conv2D(filters= 256, kernel size= 5, strides= 2)
- LeakyReLU($\alpha = 0.2$)
- Dense(units= 1, activation = 'sigmoid')

## E.3 Skin-cancer MNIST

### E.3.1 Generator layers

- Dense(units=$4 \times 4 \times 512$)
- Reshape(target shape= $(4, 4, 512)$)
- Conv2DTranspose(filters= 256, kernel size= 5, strides= 2)
- ReLU()
- Conv2DTranspose(filters= 128, kernel size= 5, strides= 2)
- ReLU()
- Conv2DTranspose(filters= 64, kernel size= 5, strides= 2)
- ReLU()
- Conv2DTranspose(filters= 3, kernel size= 5, strides= 2,activation = 'tanh')

### E.3.2 Discriminator layers

- Conv2D(filters= 64, kernel size= 5, strides= 2)
- Conv2D(filters= 128, kernel size= 5, strides= 2)
- LeakyReLU($\alpha = 0.2$)
- Conv2D(filters= 128, kernel size= 5, strides= 2)
- LeakyReLU($\alpha = 0.2$)
- Conv2D(filters= 256, kernel size= 5, strides= 2)
- LeakyReLU($\alpha = 0.2$)
- Dense(units= 1, activation = 'sigmoid')

### E.3.3 Toy binary classifier layers

- Conv2D(filters= 512, kernel size= 5, strides= 2)
- LeakyReLU($\alpha = 0.2$)
- Dense(units= 2, activation = 'relu')
- Dense(units= 1, activation = 'sigmoid')

## F Auxiliary lemmas and theorems

**Theorem 10** (Slutsky's theorem). *Let $X_n \xrightarrow{d} X$ and $Y_n \xrightarrow{d} c$, where $c$ is a constant. Then*

1. $X_n + Y_n \xrightarrow{d} X + c$;

2. $X_n Y_n \xrightarrow{d} cX$;

3. $X_n / Y_n \xrightarrow{d} X/c$ if $c \neq 0$.

**Corollary 1** (Slutsky's theorem). *Let $X_n \xrightarrow{p} c_0$ and $Y_n \xrightarrow{p} c_1$, where $c_0$ and $c_1$ are constants. Then*

1. $X_n + Y_n \xrightarrow{p} c_0 + c_1$;

2. $X_n Y_n \xrightarrow{p} c_0 c_1$;

3. $X_n / Y_n \xrightarrow{p} c_0 / c_1$ if $c_1 \neq 0$.

*Proof.* When $c$ is a constant, $Z_n \xrightarrow{p} c$ is equivalent to $Z_n \xrightarrow{d} c$. Thus we have $X_n \xrightarrow{d} c_0$ and $Y_n \xrightarrow{d} c_1$. By applying Theorem 10 (1), we have

$$X_n + Y_n \xrightarrow{d} c_0 + c_1.$$

Since $c_0 + c_1$ is a constant, it follows

$$X_n + Y_n \xrightarrow{p} c_0 + c_1.$$

Similarly, we can prove 2 and 3. $\square$

**Theorem 11** (Continuous mapping theorem). *Let $f : \mathbb{R}^m \to \mathbb{R}^q$ be a measuarable function. Define*

$$C_f = \{x : f \text{ is continuous at } x\}.$$

*If $X_n \xrightarrow{p} X$ and $\mathbb{P}(X \in C_f) = 1$, then*

$$f(X_n) \xrightarrow{p} f(X).$$

**Theorem 12** (Devroye & Gyorfi (1985)). *Let $p_N$ be an automatic kernel estimate with arbitary density $K$, as defined in Equation (17). If $h + (nh^d)^{-1} \to 0$ completely (almost surely, in probability), then $\int |p_N - p| \to 0$ (almost surely, in probability), for all density $p$ on $\mathbb{R}^d$.*

**Theorem 13** (McDiarmid's inequality). *Let $f : \mathcal{Z}^m \to \mathbb{R}$ be a function satisfying*

$$|f(z_1, \ldots, z_i, \ldots, z_m) - f(z_1, \ldots, z_i^{'}, \ldots, z_m)| \le c_i$$
$$(\forall i, \forall z_1 \ldots z_m, z_i^{'} \in \mathcal{Z}).$$

*Denote*

$$v = \frac{1}{4} \sum_{i=1}^{m} c_i^2.$$

*Let $Z_1, \cdots, Z_m$ be independent variables with support on $\mathcal{Z}$. Then*

$$\mathbb{P}(f(Z_1, \cdots, Z_m) - \mathbb{E}[f(Z_1, \cdots, Z_m)] \ge t) \le \exp\left(-t^2/(2v)\right)$$

*and*

$$\mathbb{P}(f(Z_1, \cdots, Z_m) - \mathbb{E}[f(Z_1, \cdots, Z_m)] \le -t) \le \exp\left(-t^2/(2v)\right).$$

**Lemma 4** (Chen (2017)). *With probability $(1 - \delta/2)$, we have*

$$r(x) \in [\widehat{r}_{\text{lower}}(x), \; \widehat{r}_{\text{upper}}(x)], \tag{31}$$

*where*

$$\widehat{r}_{\text{lower}}(x) = \widehat{r}_N(x) - z_{1-\delta/4}\sqrt{\frac{\mu_K \cdot \widehat{r}_N(x)}{Nh^d}},$$
$$\widehat{r}_{\text{upper}}(x) = \widehat{r}_N(x) + z_{1-\delta/4}\sqrt{\frac{\mu_K \cdot \widehat{r}_N(x)}{Nh^d}}.$$

*and $\mu_K := \int K^2(x)dx$, $z_{1-\delta/4}$ is the $(1 - \delta/4)$ quantile of a standard normal distribution.*

## G Connection to differential privacy

As shown in Theorem 1, both the optimal membership advantage and the individual privacy risk are bounded by differential privacy guarantees. We construct a toy dataset from MNIST by forming a new imbalanced dataset with 6900 digit zeros and 700 digit sixes. This dataset is also used for anomaly detection (Bandaragoda et al., 2014). We set $p = 0.5$ for simplicity. Figure 5 shows the optimal membership advantage of the DP-cGAN (Torkzadehmahani et al., 2019) with different choices of the privacy budget $\varepsilon$. As we can see here, the theoretic upper bound given by $\tanh(\varepsilon/2)$ is much larger than the estimated optimal membership advantage. Figure 6 shows the individual privacy risks for both $\varepsilon = 2$ and $\varepsilon = 10$. As expected, even the highest individual privacy risk is strictly bounded by the upper bound derived from the privacy budget $\varepsilon$. The upper bound ($tanh(\frac{2}{2}) = 0.762$ for $\epsilon = 2$, and $tanh(\frac{10}{2}) = 0.999$ for $\varepsilon = 10$). These demonstrations seem to indicate that if membership privacy is desired, using differentially private methods can lead to far too conservative models (which may lead to poorer model utility). However, it may also be that the privacy accounting in DP-cGAN is loose, thereby leading to an overestimation of $\varepsilon$.

## H The optimal membership advantage as a function of synthetic dataset size

To explore the effect the size of synthetic datasets on the membership advantage, we first trained a JS-GAN on the skin-cancer MNIST dataset. The JS-GAN was trained for 2000 epochs and several synthetic datasets of sizes varying from 10 to $10^5$ samples was generated. The adversary used was the one described in Equation 2. It can be seen in Figure 7 that as the synthetic dataset size increases, so does the optimal membership advantage.

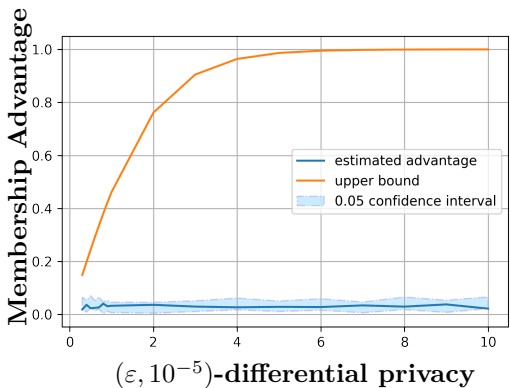

Figure 5: The optimal membership advantage and its upper bound estimated versus the privacy budget $\varepsilon$ for DP-cGANs.

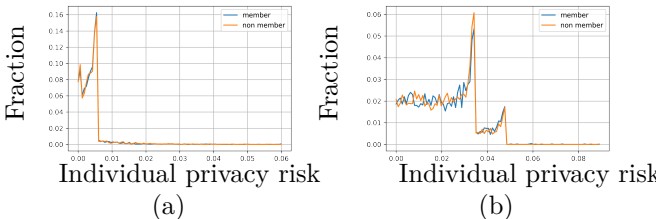

Figure 6: The individual privacy risks for DP-cGAN with privacy budget $(\varepsilon, 10^{-5})$ (a) individual privacy risks for $\varepsilon = 2$ (b)individual privacy risks for $\varepsilon = 10$.

# I   Estimation of optimal membership advantage for discriminative models

To demonstrate that our estimators of optimal membership advantage are also applicable to discriminative models, we trained a simple binary classifier (architecture described in section E.3.3) on the skin-cancer MNIST dataset (same experimental setting as the generative model). We chose two queries: i) a black-box query - the final output of the classifier, ii) a white-box query - the output of the penultimate layer of the classifier. We discretized the outputs and used our discrete estimator to estimate the optimal membership advantage in each case, as seen in Figure 8. As expected, the white-box query has a higher optimal membership advantage than the black-box query.

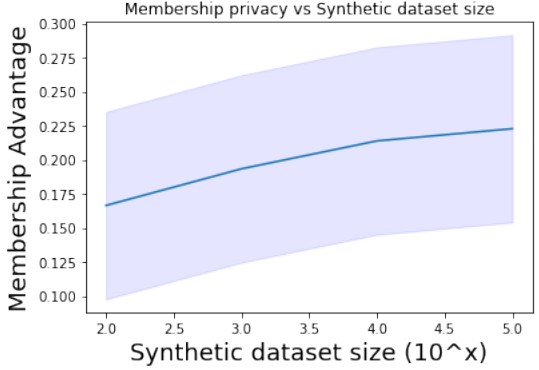

Figure 7: The optimal membership advantage vs. the synthetic dataset size.

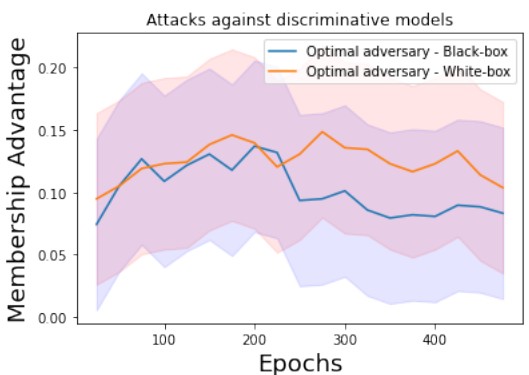

Figure 8: Estimation of the optimal membership advantage with white-box and black-box queries against a binary classification model on skin-cancer MNIST.

