# OpenReview forum: "Mace: A flexible framework for membership privacy estimation in generative models"
_TMLR — Accepted by TMLR_

### Review · Reviewer_p24z · 2022-08-09

**Summary Of Contributions:**

This paper introduces a framework for estimating membership privacy in generative models (e.g. GANs). The paper shared some of the setups with prior works such as Yeom et al 2018, and extended with more flexible assumptions to be applied in more general settings. Specifically this paper addresses the imbalance issue in prior membership privacy frameworks by utilizing a general metric class studied in Koyejo et al 2014. Then the authors made a novel connection between individual-level and population-level membership privacy. Finally the authors proposed another setting with looser assumptions than Yeom et al where only a random sample is accessible instead of the data distribution and provided practical estimators for membership risks  in this setting. The framework was evaluated on three benchmark image datasets (MNIST, CIFAR10 and skin-cancer MNIST) and on different GAN architectures.


**Broader Impact Concerns:**

No concern on the ethical implications

**Requested Changes:**

1. In Algorithm 2, the authors proposed to estimate $\eta_(Q_S(z))$ by the first set of samples, while no further explanation is provided. Would appreciate the author to elaborate more on this estimation or provide a reference.
2. In Section 6.2.1, the authors stated “our optimal membership advantage estimates are higher than the SOTA heuristic adversaries in both settings.” However, I could only find comparison with SOTA heuristics in Figure 1a for 1 setting instead of both settings. Would appreciate the authors to complete the plots for the statement.
3. In Section 6.2.2, the authors stated “In this case, the performance of the optimal adversary is consistently better than the heuristic adversary.” This is the case for the MNIST AM metric plot but not very obvious in the CIFAR10 plots where the two curves overlap from epoch 5,000 or so. Would appreciate the authors to reword or explain the CIFAR10 plot in more detail.
4. In Figure 2a, why does the 2d query curve (red) fluctuate regularly? E.g. membership advantage could drop from 0.8 to 0.3. Would appreciate authors to explain this behavior.
5. Seems like for one comparison setting, one model and one dataset are used. The paper could be strengthened by showing the comparison on all models and all datasets.

Minor and questions:
1. In Section 2.4, An adversary -> an adversary
2. All the plots in Section 6 are small. Bigger plots might be easier to read.
3. In algorithm 2, do the three sampled sets need to be disjoint? How would this impact the estimation of optimal membership advantage?
4. As the authors acknowledged in the conclusion that the framework could be extended to discriminative models. I also did not find that the framework specific to generative models, yet the title and motivation all focused on generative models. Why not emphasize that the framework is suitable for both discriminative and generative models, which covers a wider range of ML and could be more impactful.


**Strengths And Weaknesses:**

Strengths

1. The paper is well-written and organized, and it clearly stated its contribution and differences to prior works.
2. The paper is technically sound where the proposed theoretical framework for membership privacy provides a convincing lower bound of risks.
3. The framework is very flexible and can be used for multiple settings and purposes. The individual privacy risk estimation is particularly important for us to understand the privacy of the vulnerable samples.
4. The empirical evaluation demonstrated its usability and how the framework provides a tighter privacy estimation than heuristic adversaries.

Weaknesses

1. The datasets and models in the experiments are all small scaled. This is acceptable given the main contributions in the paper are in the theoretical aspects.
2. The individual risk estimation is very interesting and potentially important, however I did not find the discussion or experiments on this to be illustrative enough. Perhaps a distribution of the individual risks or more examples of the vulnerable samples would be helpful to highlight their characteristics.  Also the observation is closely related to “memorization” in neural networks discussed in Feldman 2020, which is not cited.
3. Although the authors stated the connection to differential privacy in Theorem 1, there are no empirical experiments showing this connection. It would be very helpful to show a plot similar to Figure 1 in Erlingsson et al 2020.

References:
1. Ulfar Erlingsson, Ilya Mironov, Ananth Raghunathan, Shuang Song. That which we call private. 2020.
2. Vitaly Feldman. Does Learning Require Memorization? A Short Tale about a Long Tail. 2020.

---

> ### Author Response · Authors · 2022-09-02
> **Response to Reviewer p24z**
>
> Thank you for your constructive comments and feedback.
>
> 1. We appreciate your constructive suggestion to extend the experiments for individual risk estimation. We would add the distributions of individual privacy risks of corrupted and uncorrupted images to Section 6.
>
> 2. We have added an experiment to compare the upper bound of the membership advantage for a DP algorithm and its optimal membership advantage to show the connection between differential privacy and membership advantage to Appendix G.
>
> 3. For Algorithm 2, we followed Koyejo et al. (2014) to have disjoint sets to estimate $\eta$ and $t_\ell$, respectively. The main reason is to meet the requirement of independence of random variables. This is essential to bound the difference between membership advantage and its empirical estimate by Hoeffding's inequality as in Koyejo et al. (2014). We have added Theorem 1 (Koyejo et al. (2014)) to Section 5.2 to better motivate Algorithm 2. We show that the membership advantage of ${\rm sgn}(\eta(Q_S(z))-t_\ell)$ (defined in Algorithm 2) converges to the optimal membership advantage in probability under some specific condition.
>
> 4. Motivated by your comments about the estimate of $\eta$, we proved Theorem 7 for the case when $t_l$ is known: if $E|\hat{\eta}-\eta|^\sigma\to 0$ in probability for some $\sigma\ge 1$, the membership advantage of ${\rm sgn}(\hat{\eta}(Q_S(z))-t_\ell)$ (defined in Algorithm 2) converges to the optimal membership advantage in probability under some specific condition. Hopefully this would address your concerns about how to estimate $\eta$ in Algorithm 2. Using a suitable strongly proper loss function, we can obtain an estimator $\hat{\eta}$ satisfying $E|\hat{\eta}-\eta|^2\to 0$ by the proof of Theorem 5 (Menon et al. (2013)). We further split up the original Algorithm 2 into two algorithms depending on whether $t_\ell$ is known to make it easier to follow.
>
> 5. In Section 6.2.1, we have added the other graph for the query against the generator to Figure 1, as suggested.
>
> 6. For the CIFAR10 plot in Section 6.2.2, we have reworded our description. Also, it might be because of our choice of the method to estimate $\eta$. Our current method first estimates $P(Q_S(z)|m=1)$ and $P(Q_S(z)|m=-1)$ using the frequency-based plug-in method as described in Section 4.1. Then plug-in $\eta=\mathbb{P}(Q_S(z)|m=1)p/( \mathbb{P}(Q_S(z)|m=1)p+ \mathbb{P}(Q_S(z)|m=-1)(1-p))$. Adapting Menon et al. (2013) and Koyejo et al. (2014), we would estimate $\eta$ using the regularized logistic regression to fulfill the condition of Theorem 5 (Menon et al. (2013)), and update Figure 3.
>
> 7. Regarding the fluctuation in Figure 2a, it is an artifact of privGAN training. privGAN split the training data into two subsets in the first place, and train two pairs of generator and discriminator. 2d used the output of the two discriminators, and that might be the reason why the curve is noisier.
>
> 8. In Section 6, we specifically design each experiment to demonstrate the usability of MACE. For example, we showed how to estimate the optimal membership advantage under the accuracy-based metric with regard to different query types on the CIFAR-10 dataset for the WGAN-GP model. We demonstrated how to apply MACE under the generalized metric using experiments on the MNIST and CIFAR-10 datasets for the WGAN-GP model. In total we utilized 3 GAN models (WGAN, JS-GAN, privGAN) applied to 4 datasets (MNIST, CIFAR-10, skin-cancer MNIST and perturbed skin-cancer MNIST). Furthermore, we have a discriminative CNN model applied to skin-cancer MNIST. Given that our paper is primarily aimed at introducing a new theoretical framework, we believe that we have performed more experimental comparisons compared to similar papers e.g. Yeom et. Al. (2018). Additionally, as mentioned in point 2, we will add a differentially private model (DP-GAN) to demonstrate the connection of our framework to differential privacy.
>
> 9. We agree with you that the framework is not specific to generative models. It can apply to other settings such as discriminative models. However, the main motivation behind this work is to estimate the membership privacy risks for the generative models in order to provide a membership privacy certificate for synthetic data sharing.
>
> 10. We have fixed all the typos, cited the relevant work, and made plots in Section 6 larger.
>
> Koyejo et al. (2014):  Koyejo, Oluwasanmi O., et al. "Consistent binary classification with generalized performance metrics." Advances in neural information processing systems 27 (2014).
>
> Menon et al. (2013): Menon, Aditya, et al. "On the statistical consistency of algorithms for binary classification under class imbalance." International Conference on Machine Learning. PMLR, 2013.
>
> Yeom et. Al. (2018): Yeom, Samuel, et al. "Privacy risk in machine learning: Analyzing the connection to overfitting." 2018 IEEE 31st computer security foundations symposium (CSF). IEEE, 2018.

---

### Review · Reviewer_3Yd4 · 2022-08-14

**Summary Of Contributions:**

The main contribution of the paper is to develop a framework to estimate the risk of membership inference attacks when the adversary is given oracle access to the model. The framework in the paper is general in the sense that it extends to general risk class considered in previous works and can measure individual level membership privacy risks,

**Broader Impact Concerns:**

N/!

**Requested Changes:**

Please take a look at the weakness section above.

**Strengths And Weaknesses:**

The entire use of the framework allows the authors to look at different studies of membership inference attacks using the same lens. Furthermore, it allows them to go beyond the previous studies by considering individual level privacy risks as well as when one does not have access to the entire training data; the latter makes it a more practical attack because it is not often the case that we have access to te entire training data.

I have a few questions. Reading through your definition of the adversarial model, it seems like it is not restricted to GAN. It resembles a lot of what we usually see in cryptography. In cryptography, one would usually write such definitions as an adversarial game where the adversary is given auxiliary inputs. From the definition of the paper, I do not see that to be the case. Is it by choice? If not, then clarify. If there is auxiliary input available to the adversary, then mention it. Also, is the adversary passive or active, adaptive, or non-adaptive, honest-but-curious or malicious? None of that is clear in the paper.

For the weakness, I found the writing (especially, the proofs) very sloppy. This makes it hard to verify the correctness of the paper. A journal paper should ideally have a much higher editor quality than a conference paper and I feel the paper falls short of that. I will use this opportunity to give some suggestions so that the authors can submit a version that I can verify before pointing to specific instances:
1. Please be consistent in the use of different citation techniques available in natbib. If you are saying, Following Chen et al. 2018, maybe, use the one that does not put parenthesis to be consistent with the later usage.
2. If you are using two or more items, then separate them with a conjunction. Like end of page 3, (1) <text>; and (2).
3. Use space between phrases and citations. Like, Hayes et al., 2019 assumes. assumes -> assume.
4. Change the phrase We then to We next or something like that. It is a very repetitive choice of phrase.
5. Please use active voice instead of passive voice. One example is line 3 and 6 in the description of Algorithm 1.

Some specific instances:
1. After Definition 3. we -> We
2. Do not start a new sentence with a conjunction. A specific example is right after equation (17).
3. What is N in theorem statements? Define its value. I am guessing it is N=N_1+N_2 from Experiment 2. A theorem statement should be self-contained.
4. Which theorem belongs to Algorithm 1?
5. Algorithm 2: the line numbers are missing.
6. Appendix D.4: What is p? Is p(\cdot) a function? It seems like the authors have overloaded the symbol p and it makes the reviewer's job very difficult. The same issue is in Appendix D.4.
7. AdvI should be in math font to be consistent.
8. FP, etc are said to be that they are conditional forms, and the expression is given later. I would rather have the mathematical expression first before the description of what it means.
9. Proof of Theorem 4: I would elaborate on all the steps in the proof. There is no page limit and every mathematical expression should be spelled out. It might be helpful to readers if the authors do not write the mathematical expression as a paragraph but in the math environment.
10. \hat p tends in probability to the expression is only used in Theorem 2's proof unless the authors have overloaded p_N to denote p.
11. Please use a preliminaries section to state the version of Slutsky's theorem and the continuous mapping theorem that you use. It is very important given the repeated use of the aforementioned theorems.
12. Elaborate on how 1 follows by Slutsky theorem and the third condition.
13. Why is 2p/N_1 = 2/N? Again, what is N?
14. I cannot find Theorem 7.

---

> ### Author Response · Authors · 2022-09-02
> **Response to Reviewer 3Yd4**
>
> Thank you for your constructive comments and feedback.
>
> 1. We agree that the experiment is similar to a cryptographic game where the adversary can observe the output of the query function. We do not assume the adversary has access to any additional auxiliary input apart from the output of the query function. But such an adversary can be modeled in our experiment as well. The query input is the challenge point ‘z’ chosen from the training dataset or the remaining distribution by the challenger based on probability ‘p’. The adversary is passive, non-adaptive and as well as honest-but-curious. The adversary does not manipulate the training process of the model or tamper with the training dataset or with the challenge point. The adversary makes an independent guess for the membership of each of the challenge point ‘z’ and hence is in a non-adaptive setting.
>
> 2. In theorem statements, N=N_1+N_2 from Experiment 2. We added the definition of N to the theorems in the revision.
>
> 3. Theorems 5 and 6 belong to Algorithm 1.
>
> 4. p in Appendix D.4 means the prior membership probability as defined in Experiment 1 and 2. $p(\cdot)$ or $p(x)=\mathbb{P}(Q_S(z)=x|m=1)$ is a function. It was defined in the first paragraph in Section 4.1.2. We regret choosing notations that caused confusion. In the revision, we instead denote $\mathbb{P}(Q_S(z)=x|m=1)$ and $\mathbb{P}(Q_S(z)=j|m=1)$ as $r(x)$ and $r_j$, respectively.
>
> 5. We have elaborated on the steps in the proof of Theorem 4.
>
> 6. In Theorem 2, $\hat{p}_j$ is an estimate of $p(Q_S(z)=j)=\mathbb{P}(Q_S(z)=j|m=1)$ for the discrete case. In Theorem 3, we use $p_N$ to denote the kernel density estimator of $p(x)=\mathbb{P}(Q_S(z)=x|m=1)$ for the continuous case.
>
> 7. We have added Slutsky's theorem and the continuous mapping theorem to Appendix F. Auxiliary lemmas and theorems.
>
> 8. The reason why $2p/N_1 = 2/N$ is as follows: N is the total number of samples sampled by Experiment 2. N1 is the number of samples drawn from the training set S. We assumed $N_1=N*p$ in Experiment 2.  Thus, $2p/N_1=2/N$.
>
> 9. Theorem 7 is in Appendix F. Auxiliary lemmas and theorems.
>
> 10. We thank the review for the careful reading and valuable suggestions. We have fixed all the typos and minor writing issues that have been kindly pointed out.

---

> > ### Comment · Reviewer_3Yd4 · 2022-09-19
> > **Response and revision**
> >
> > I thank the authors for making the changes. The paper reads much better now. I apologize for the delay but I wanted to be thorough in ensuring that all the changes are in place.
> >
> > A small suggestion: If you make changes that you wish to highlight, it might be a good idea to make it in a different color. It helps the reviewers and helps you in getting the response back faster.

---

> > > ### Author Response · Authors · 2022-09-19
> > > **Re: Response and revision**
> > >
> > > Thanks for the suggestion! I have just uploaded a revision with major changes highlighted in blude.

---

### Review · Reviewer_6r1B · 2022-08-20

**Summary Of Contributions:**

This paper studies the problem of membership privacy estimation in generative models. More specifically, the authors propose a framework to estimate the maximum membership privacy risk for generative models. The proposed framework is more general than the previous heuristic methods and is able to characterize the individual-level membership privacy risk. Experiments validate the advantage of the proposed method.

**Requested Changes:**

My main concerns of the current paper are as follows:
1. What is the motivation of your proposed query function in equation (2)? What is the advantage of the proposed query function compared with the previous one?
2. Is there a scaling issue in equation (4)? What is the output range of $D$ and $d$?
3. In section 3.1, the Adv definition in equation (1) is based on $p=0.5$ in Experiment 1. Why can it be used to derive the proposed optimal advantage since you are considering a general $p$?
4. In Experiment 2, your method also needs access to the distribution D, and why is this better than the previous work, e.g., Yeom et al. and Jayaraman et al.?


**Strengths And Weaknesses:**

Strengths:
1. The proposed framework is very general and can be used to estimate privacy risk under different assumptions, i.e., releasing a model or synthetic dataset.
2. The proposed method takes into account the unbalanced membership in the data.
3. It can be used to estimate individual-level risk.
4. Experiments show the effectiveness of the proposed framework.

Weakness:
1. The motivation of the proposed query function is not clear.
2. The accuracy based metric need to be further explained.

---

> ### Author Response · Authors · 2022-09-02
> **Response to Reviewer 6r1B**
>
> Thank you for your constructive comments and feedback.
>
> 1. The motivation for the proposed query function in equation (2) is that the generator usually memorizes the training data thus it generates synthetic dataset close to the training data. Under this assumption, if a sample x is closer to the synthetic dataset, it is more likely that x belongs to the training data. The motivation of the query function in equation (1) is that a sample x would be more likely a training sample if the probability that a sample g generated by the generator belongs to the $\epsilon$-neighborhood of x is increased. The advantage of the proposed query function in equation (1) is its simplicity: we must tune the hyperparameter $\epsilon$ for the query function in equation (1), as outlined in Hilprecht et al. (2019). In this paper, we focus on the formulation of the membership privacy risk given a query access. While we provided a representative query for different scenarios, the selection of the optimal query function is beyond the scope of this paper.
>
> 2. The output range of D is [0,1]. d is the Euclidean distance, and its output range is $[0,\infty]$ in theory. Though it is not a problem of our experiments, a re-scaling would be suggested if the empirical distance range differs too much from [0,1].
>
> 3. The original Adv definition (Yeom et al. 2018) was based on p=0.5 in Experiment 1, and we adapted it to the case where p is not equal to 0.5. In equation (10), the membership advantage is defined as 2 times the expected accuracy of the adversary’s prediction minus 1 no matter whether p equals to 0.5 or not.
>
> 4. Previous works require full access to the distribution D i.e. the probability density function for a continuous variable or probability mass function for a discrete variable. As a comparison, our framework can still be applicable when one can only access the training set S and the data distribution D via a simple random sample even without full access to S and D, as described in Experiment 2.
>
> 5. To better motivate the proposed query, we added the following paragraph in Section 2.2.1:
> Similar to these approaches, we assume that the generator memorizes the training data thus it generates synthetic dataset close to the training data. Under this assumption, if a sample x is closer to the synthetic dataset, it is more likely that x belongs to the training data.  Hence for a sample z, we consider using the nearest neighbor distance to synthetic datasets as the query function:
>
> 6. To better explain the accuracy-based metric, we added the following sentence in front of Def 1:
> We define the membership advantage of the query $Q_S$ by an adversary $ {\cal A}$ as the rescaled expected accuracy of the membership inference attack adversary $ {\cal A}$. When p=0.5, the membership advantage is equal to the difference between the adversary’s true and false positive rates.
>
> Reference:
>
> Hilprecht et al. (2019): Hilprecht, Benjamin, Martin Härterich, and Daniel Bernau. "Monte Carlo and Reconstruction Membership Inference Attacks against Generative Models." Proc. Priv. Enhancing Technol. 2019.4 (2019): 232-249.
>
> Yeom et al. 2018: Yeom, Samuel, et al. "Privacy risk in machine learning: Analyzing the connection to overfitting." 2018 IEEE 31st computer security foundations symposium (CSF). IEEE, 2018.

---

### Comment · Action_Editors · 2022-09-07
**Revision?**

Thanks authors for your comments and responses to the reviewers. Since OpenReview offers the option of submitting revisions, do you plan to upload a revision of the paper with the changes that you have promised to the reviewers? And potentially clarifying the points of confusion, if appropriate.

---

> ### Author Response · Authors · 2022-09-07
> **Manuscript revision**
>
> We thank the action editors for the suggestion! We are working on the revision and would expect to submit it in two weeks. Let us know if it works or not.

---

> > ### Comment · Action_Editors · 2022-09-08
> > **Deadline**
> >
> > The official recommendations are able to be submitted by reviewers any time between now and September 17. I would say the earlier the better, so that the reviewers would be able to take them into account. Most authors have submitted their revision at the same time as the responses to reviewers.

---

> > > ### Author Response · Authors · 2022-09-08
> > > **Re: Deadline**
> > >
> > > Thanks for your suggestions! We would submit as soon as possible before Sept 17.

---

### Decision · Action_Editors · 2022-09-26

**Recommendation:** Accept as is

**Comment:**

This paper makes a contribution in the area of estimating privacy of generative models. The reviewers initially had some feedback and comments on aspects of the work, which the authors addressed in an updated revision. Beyond this, there was little additional discussion among the reviewers, who felt their questions and concerns were addressed by the authors, and thus deemed this paper ready for publication.

---

> ### Author Response · Authors · 2022-10-06
> **Thank you!**
>
> We would like to thank all the action editors and reviewers for their constructive suggestions. Thanks to your valuable feedbacks, our work has been greatly improved compared to our original submission. Thanks again!